# Sensitivity of future Continental United States water deficit projections to General Circulation Model, evapotranspiration estimation method, and greenhouse gas emission scenario

**S. Chang[1], W. Graham[1, 2], S. Hwang[3], and R. Muñoz-Carpena[4]**

[1] Department of Agricultural and Biological Engineering, University of Florida, 570 Weil Hall, PO Box 116601, Gainesville, FL 32611, USA

[2] Water Institute, University of Florida, 570 Weil Hall, PO Box 116601, Gainesville, FL 32611, USA

[3] Department of Agricultural Engineering, Institute of Agriculture and Life Science, Gyeongsang National University, Jinju, 660-701, South Korea

[4] Department of Agricultural and Biological Engineering, University of Florida, 287 Frazier Rogers Hall, PO Box 110570, Gainesville, FL 32611, USA

Correspondence to: W. Graham (wgraham@ufl.edu)

## Abstract

Projecting water deficit under various possible future climate scenarios depends on the choice of General Circulation Model (GCM), reference evapotranspiration ($ET_0$) estimation method and Representative Concentration Pathway (RCP) trajectory. The relative contribution of each of these factors must be evaluated in order to choose an appropriate ensemble of future scenarios for water resources planning. In this study variance-based global sensitivity analysis and Monte Carlo filtering were used to evaluate the relative sensitivity of projected changes in precipitation (P), $ET_0$ and water deficit (defined here as $P - ET_0$) to choice of GCM, $ET_0$ estimation method and RCP trajectory over the continental United States (US) for two distinct future periods: 2030-2060 (future period 1) and 2070-2100 (future period 2). A total of 9 GCMs, 10 $ET_0$ methods and 3 RCP trajectories were used to quantify the range of future projections and

estimate the relative sensitivity of future projections to each of these factors. In general, for all
regions of the Continental US, changes in future precipitation are most sensitive to the choice of
GCM, while changes in future $ET_0$ are most sensitive to the choice of $ET_0$ estimation method.
For changes in future water deficit, the choice of GCM is the most influential factor in the cool
season (Dec – Mar) and the choice of $ET_0$ estimation method is most important in the warm
season (May – Oct) for all regions except the South East US where GCM and $ET_0$ have
approximately equal influence throughout most of the year. Although the choice of RCP
trajectory is generally less important than the choice of GCM or $ET_0$ method, the impact of RCP
trajectory increases in future period 2 over future period 1 for all factors. Monte Carlo filtering
results indicate that particular GCMs and $ET_0$ methods drive the projection of wetter or drier
future conditions much more than RCP trajectory; however the set of GCMs and $ET_0$ methods
that produce wetter or drier projections varies substantially by region. Results of this study
indicate that, in addition to using an ensemble of GCMs and several RCP trajectories, a range of
regionally-relevant $ET_0$ estimation methods should be used to develop a robust range of future
conditions for water resource planning under climate change.

## 1. Introduction

Climate change will result in significant impacts on hydrologic processes. The 2014 Fifth
Assessment Report (AR5) of the Intergovernmental Panel on Climate Change (IPCC) reported
that climate change will significantly affect future precipitation (P), temperature (T) and
reference evapotranspiration ($ET_0$) and these changes will affect the quantity and quality of water
resources. The most recent report of the National Climate Assessment and Development
Advisory Committee (NCADAC, 2013) indicated that the average annual temperature in the
United States (US) has increased by 0.7 °C to 0.9 °C since record keeping began in 1895 and is
expected to continue to rise (Georgakakos et al., 2014; Walsh et al., 2014). The NCADAC report
also indicated that Coupled Model Intercomparison Project 5 (CMIP5) General Circulation
Model (GCM) precipitation projections show a consistent increase in Alaska and the far north of
the continental US and a consistent decrease in the far Southwest US, but that GCM projections
are inconsistent in the precipitation transition zone of the US continent. The uncertainty in
climate change projections makes actionable water resources planning difficult in many regions.
In order to predict changes in the hydrologic cycle, and future water supply and demand,
estimates of changes in P, T and $ET_0$ must be evaluated on a regional basis, and the uncertainty
of these estimates must be quantified (Ishak et al., 2010).

Previous research has evaluated existing and potential future spatiotemporal changes in P,

T and $ET_0$ for various regions around the globe (e.g. Chaouche et al., 2010; Chong-Hai and
Ying, 2012; Johnson and Sharma, 2009; Kharin et al., 2013; Maurer and Hidalgo, 2008;
Quintana Seguí et al., 2010; Sung et al., 2012; Thomas, 2000; Wang et al., 2013; Xu et al.,
2006). It is well known that future GCM projections of temperature and precipitation vary
significantly due to both the different radiative forcing assumptions of carbon dioxide scenarios
(e.g. CMIP3 Special Report on Emissions Scenarios (SRES) and CMIP5 Representative
Concentration Pathways (RCP trajectories)) and different GCM model physics (Hawkins and
Sutton, 2009, 2010). Future $ET_0$ projections have been shown to depend on $ET_0$ estimation
methods in addition to GCMs. For example, Wang et al. (2015) used projections from the
CMIP3 HADCM3 model A2 scenario and found that the physically-based Penman-Monteith
equation, which uses less reliable GCM projection data (including vapor pressure and solar
radiation), and the empirical temperature-based Hargreaves equation showed similar patterns but
different magnitudes for future $ET_0$ changes over the Hanjiang River Basin in China. Kingston et
al. (2009) used 5 GCMs from the CMIP3 climate projections and 6 different $ET_0$ equations to
estimate global $ET_0$ and found that the choice of $ET_0$ method contributes to different projections
of the future state of water resources which varies by region. They found that the Hamon and
Jensen-Haise $ET_0$ estimates showed the greatest changes in both humid and arid regions while
the Penman-Monteith and Priestley-Taylor estimates frequently showed smallest change.
Similarly McAfee (2013) used three $ET_0$ equations with 17 CMIP3 GCMs to evaluate the
uncertainty of future global $ET_0$ projections and found that the Hamon equation showed more
significant and consistently positive trends in $ET_0$ compared to the Priestley-Taylor and Penman
methods.

Models developed to estimate future water supply and demand as a result of projected

climate change use many different types of $ET_0$ estimation methods (Zhao et al., 2013). Because
the choice of $ET_0$ estimation method may be as important as the choice of GCM or RCP
trajectory, better understanding of the contribution of each of these factors to the overall
prediction uncertainty of future water availability or water deficit is necessary (Taylor et al.,
2013).  Kay and Davies (2008) compared the performance of the Penman-Monteith equation and
a simple temperature-based $ET_0$ method using climate data from five global and eight regional
climate models over Britain. They found that the two methods showed very different changes in
$ET_0$ for the period 2071-2100 under the A2 emission scenario, and different flow predictions for
three catchments when the data were used to force a rainfall-runoff model. Kay and Davies
results suggest that hydrological prediction uncertainty due to $ET_0$ formulation was smaller than
that due to GCM structure or RCM structure for their study region.  Bae et al. (2011) evaluated
the uncertainty contributed by choice of GCM and hydrologic model for the Chungju Dam basin,
Korea. They found that hydrologic model structural differences contributed greater uncertainty
than GCM selection to winter runoff prediction. Koedyk and Kingston (2016) found that for the
Waikaia River, New Zealand $ET_0$ method contributed more uncertainty than GCM selection
when predicting $ET_0$, but that runoff predictions were more sensitive to GCMs than to $ET_0$
methods. Thompson et al. (2014) evaluated the effect of using different GCMs and different $ET_0$
methods on discharge predictions for the Mekong River in Southeast Asia and found that GCM-
related uncertainty was greater than the $ET_0$ method related uncertainty.

In this study we perform a comprehensive evaluation of the relative sensitivity of future

P, $ET_0$ and water deficit (defined here as P- $ET_0$) projections to choice of GCM, $ET_0$ method and
RCP trajectory over the continental USA using CMIP5 GCM model outputs to provide new
insights that will inform more robust future water resource planning efforts. Variance-based
global sensitivity analysis (Saltelli et al., 2010) and Monte Carlo Filtering (Rose et al., 1991) are
used to quantify the uncertainty and important input factors controlling these projections. Global
sensitivity analysis (GSA) apportions the total output uncertainty simultaneously onto all the
uncertain input factors described by marginal probability density functions, and thus is preferred
over the local, one factor at a time, sensitivity analyses that have been previously reported
(Homma and Saltelli, 1996; Saltelli, 1999). Monte Carlo Filtering can identify sets of model
simulations and input factors that meet a specified criteria or threshold. Thus global sensitivity
analysis and Monte Carlo Filtering offer an opportunity to gain insight into the sources of
uncertainty, and drivers of particular types of wet/dry behavior, when estimating future water
deficit under projected climate change.

## 2. Methods

All retrospective and future climate variables were obtained from the CMIP5 archive
(accessible for download at http://pcmdi9.llnl.gov/). The "historical" runs of CMIP5 were used
for the retrospective period (1950-2005) and the same ensemble member runs (r1i1p1 ensemble)
of CMIP5 were used for two future periods: future period 1 (2030-2060), and future period 2
(2070-2100). Data for three RCP trajectories, RCP2.6, RCP4.5 and RCP8.5 were included in the
analyses. Taylor et al. (2012) described an overview of CMIP5 and RCP trajectories and
compared the differences between CMIP5 and CMIP3 model projections.
Data from the CMIP5 archive were used to calculate monthly mean P, $ET_0$, and P- $ET_0$
(water deficit) for the retrospective and both future periods over each of the nine U.S. climate
regions identified by the National Climatic Data Center (Karl and Koss, 1984 (Fig. 1)). Future
changes in monthly mean P, $ET_0$, and P- $ET_0$ were estimated by subtracting the monthly mean
value for the retrospective period from the monthly mean value for future period 1 or future
period 2, as appropriate (Baker and Huang, 2014).
Ten commonly used reference evapotranspiration estimation methods (Hargreaves,
Blaney-Criddle, Hamon, Kharrufa, Irmak-Rn, Irmak-Rs, Dalton, Meyer, Penman-Monteith and
Priestley-Taylor) were used in this study. The methods can be further classified into temperature-
(Hargreaves, Blaney-Criddle, Hamon and Kharrufa), radiation (Irmak-Rn, Irmak-Rs and
Priestley-Taylor), mass transfer (Dalton and Meyer), and combination (Penman-Monteith)
equations. These equations are well-described in many papers (e.g., Allen et al., 1998;
Hargreaves and Allen, 2003; Irmak et al., 2003; Tabari, 2010; Tabari et al., 2013; Xu and Singh,
2001) and are summarized in Table 1 (hereafter precipitation is referred to as P, and reference
evapotranspiration is referred to as $ET_0$ for convenience).
Variables directly used from the CMIP5 monthly model output included precipitation (pr,
P in this study), maximum and minimum temperature (tasmax and tasmin), radiation (rlds, rlus,
rsds, and rsus), air pressure (psl and ps), and wind speed (sfcWind). The abbreviations for these
variables are as defined in the CMIP5 archive and explained in the PCMDI server (Program For
Climate Model Diagnosis and Intercomparison, http://cmip-
pcmdi.llnl.gov/cmip5/docs/standard_output.pdf). Other variables needed in the ten reference
evapotranspiration equations were calculated using the variables from CMIP5 monthly model
output (for details see Table 1). Monthly output that included all the variables needed for the
Penman-Monteith reference evapotranspiration method (the most data intensive method) was
available for both the retrospective period, and for the RCP2.6, RCP 4.5, and RCP8.5 trajectories
for the future periods, for 9 CMIP5 models. Table 2 lists the 9 models from the CMIP5 archive
that were used in this study.

The sensitivity of changes in future P, $ET_0$ and water deficit (defined here as P- $ET_0$) to

the choice of GCM, $ET_0$ estimation method, and RCP trajectory was evaluated using the
variance-based GSA method of Saltelli et al. (2010). Given a model of the form $Y =$
$f(X_1, X_2, \dots X_k)$, with $Y$ a scalar, the variance-based first order effect for a generic factor $X_i$ can
be written (Saltelli et al., 2010):
$$V_{X_i}\left(E_{X_{\sim i}}(Y|X_i)\right) \tag{1}$$
where $X_i$ is the $i$-th factor (in our case either GCM, $ET_0$ method or RCP trajectory) and $X_{\sim i}$ is the
vector of all factors except $X_i$. The expectation operator $E_{X_{\sim i}}(Y|X_i)$ indicates that the mean of
$Y$ is taken over all possible values of $X$ except $X_i$ (i.e. $X_{\sim i}$ ) while keeping $X_i$ fixed. The variance,
$V_{X_i}$, is then taken of this quantity over all possible values of $X_i$.

The first order sensitivity coefficient is expressed as:

$$S_i = \frac{V_{X_i}(E_{X_{\sim i}}(Y|X))}{V(Y)} \tag{2}$$
Where $V(Y)$ the total variance of Y over all $X_i$. $S_i$ is a normalized index varying between 0 and
1, because $V_{X_i}\left(E_{X_{\sim i}}(Y|X_i)\right)$ varies between 0 and $V(Y)$ according to the identity (Mood et al.,

1974):

$$V_{X_i}\left(E_{X_{\sim i}}(Y|X_i)\right) + E_{X_i}\left(V_{X_{\sim i}}(Y|X_i)\right) = V(Y) \tag{3}$$
As indicated above $V_{X_i}\left(E_{X_{\sim i}}(Y|X_i)\right)$ is the first order effect of $X_i$ on the model output
$Y$, while $E_{X_i}\left(V_{X_{\sim i}}(Y|X_i)\right)$ is called the residual. The total effect index, including first order and
higher order effects (i.e. interactions between factor $X_i$ and other factors) of the factor $X_i$ on the
model output is calculated (Saltelli et al., 2010):
$$S_{T_i} = \frac{E_{X_{\sim i}}(V_{X_i}(Y|X_{\sim i}))}{V(Y)} = 1 - \frac{V_{X_{\sim i}}\left(E_{X_i}(Y|X_{\sim i})\right)}{V(Y)} \tag{4}$$
The first order sensitivity of estimated future changes in mean monthly P, $ET_0$, and P-
$ET_0$ to choice of GCM, $ET_0$ estimation method and RCP trajectory were calculated over the nine
US climate regions for each future period in order to evaluate the relative contributions of each
of these factors on the uncertainty of future changes. A total of 270 simulations (9 GCMs $\times$ 10
evapotranspiration methods $\times$ 3 RCP trajectories) was used in the analysis. Sensitivity of
projected changes in P were evaluated for both choice of GCM and choice of RCP trajectory.
Sensitivity of projected changes in $ET_0$ and P- $ET_0$ were evaluated for choice of GCM, choice of
$ET_0$ estimation method, and choice of RCP trajectory.
For projected changes in water deficit (P- $ET_0$) Monte Carlo filtering (Saltelli et al., 2008)
was used to identify whether projected wetter or drier future conditions (i.e. larger or smaller
water deficit) could be attributed to specific GCMs, $ET_0$ estimation methods, or RCP trajectories.
For each future period the ensemble of 270 projections of change in water deficit were
categorized as either wet future condition (mean change in $(P - ET_0) \geq 0$) or dry future
condition (mean change in $(P - ET_0) < 0$). Next for each factor ($X_i$ =GCM, $ET_0$ method, RCP
trajectory) the histograms of wet $(f_{wet}|X_i)$ and dry $(f_{dry}|X_i)$ future conditions over the range of
possible values of that factor were estimated. To identify the factors that are most responsible for
driving the model into projected wet or dry future conditions for each factor, $X_i$, the distributions
$(f_{wet}|X_i)$ and $(f_{dry}|X_i)$ were tested for significant difference using the $X^2$ two sample test for
categorical variables with $\alpha$=0.05 (Rao and Scott, 1981). If for a given factor $X_i$ the two
distributions are significantly different, then $X_i$ is a key factor in driving into either a wet or dry
condition (Saltelli et al., 2008).
Because GCM predictions are known to contain systematic biases (Hwang and Graham,
2013; Wood et al., 2002, 2004) we evaluated the sensitivity of the mean monthly change in raw
climate predictions between retrospective and future periods to the choice of GCM, $ET_0$
estimation method and RCP trajectories. This is analogous to using the delta change GCM bias
correction method that involves shifting the mean of a series of observed climate data by the
mean difference in raw GCM output between the corresponding observed time period and the
desired future period. Teutschbein and Seibert (2012) pointed out that all bias correction methods
are based on the stationarity principle that assumes that similar biases occur in the retrospective
and future predictions and thus the same bias-correction algorithm may be applied to both.
Muerth et al. (2013) found that the impact of bias correction on the relative change of flow
indicators between retrospective and future periods was weak for most indicators, however
Pierce et al. (2015) found that some bias correction methods altered model-projected changes in
mean precipitation and temperature. LaFond et al. (2014) found that the delta change GCM bias
correction method was more useful for simulating hydrologic extreme events than the quantile
mapping bias correction method as it preserved daily climate variability better. In this study, we
differenced raw rather than bias corrected GCM outputs in order to prevent spurious alteration of
the climate change signal between retrospective and future GCMs that might be induced by the
bias correction method.

**3. Results**
3.1. Projected P, $ET_0$, and water deficit change in the $21^{st}$ century
Future P, $ET_0$ and water deficit projections include large uncertainties stemming from
different sources. Figures 2 and 3 present maps of the mean change (Fig. 2) and the standard
deviation of change (Fig. 3) in annual P (top chart), $ET_0$ (middle) and water deficit ($P - ET_0$;
bottom) over the continental US calculated over all GCMs, $ET_0$ estimation methods, and RCP
trajectories for future period 2 (2070-2100). Major portions of the West, Southwest and South
show a mean decrease in annual precipitation, while the rest of the continental US shows a mean
increase (Fig. 2 (a)). Future annual $ET_0$ shows a mean increase over retrospective annual $ET_0$
over the entire US (Fig. 2 (b)), with the largest increase in the South region. Following the
patterns of P and $ET_0$, future annual water deficit (P – $ET_0$) shows a significant mean decrease in
the West, Southwest and South regions and a slight decline, or negligible change in most other
regions (Fig. 2 (c)). These mean changes in annual P, $ET_0$ and P- $ET_0$ are relatively small
compared to the standard deviation of changes in annual P, $ET_0$, and P – $ET_0$ (Fig. 3). Water
deficit in particular has a large standard deviation, resulting in coefficients of variation larger
than one throughout the continental US. Similar results are shown in the Fig. S-1 and Fig. S-2 for
future period 1 (2030-2060) in the supplemental materials.
Figure 4 shows the seasonal changes in the monthly mean and standard deviation of
water deficit (P – $ET_0$) over the nine US regions. Blue and red lines represent the changes in
monthly mean water deficit for future period 1 and future period 2, respectively and the error
bars represent one standard deviation around each mean value. All regions of the continental US
show drier conditions (negative mean changes) in the summer season (Jun – Aug). Southern
regions (Southeast, South, Southwest and West) show drier conditions throughout the year,
however northern portions of the US (i.e. the Northeast, Ohio Valley, Upper Midwest, Northern
Rockies and Plains and Northwest) show wetter conditions (positive mean changes) in the winter
season.
3.2. Global sensitivity analysis of projected changes
Figure 5 shows the first order sensitivity of change in P to GCM and RCP trajectory over
the nine US climate regions for future periods 1 and 2. For projected changes in P, the choice of
GCM is generally more important than choice of RCP trajectory for all regions and both future
periods.  First order sensitivities of mean change in $ET_0$ to GCM, $ET_0$ method and RCP
trajectory are shown in Fig. 6. This figure clearly shows that the choice of $ET_0$ method is the
most influential factor for projecting change in $ET_0$ for both future periods, except for the month
of March in the Northeast, Upper Midwest and Northern Rockies and Plains. High sensitivity of
mean change in $ET_0$ to GCM selection occurs in spring for several regions (Northeast, Upper
Midwest and Northern Rockies and Plains), indicating a divergence of model predictions during
this time. The influence of the RCP trajectory on $ET_0$ increases in future period 2 over future
period 1, with a concomitant decrease in the influence of both $ET_0$ method and GCM. In future
period 1 the GCM sensitivity coefficients are greater than the RCP trajectory sensitivity
coefficients over most regions; however, in future period 2 the RCP sensitivity coefficients
become more important. Figure 7 shows that projected change in water deficit depend strongly
on both the choice of GCM and $ET_0$ estimation method. In all regions except the Southeast
projected change in water deficit is most sensitive to $ET_0$ estimation method in the warm season
(May through October) and most sensitive to GCM in the cool season (December through
March). For the Southeast region the sensitivity coefficients for GCM and $ET_0$ method are quite
similar throughout the year. The sensitivity coefficients for RCP trajectory are very low in future
1, but increase in future 2, becoming approximately equal to the GCM sensitivity coefficients in
the summer season in future 2.
3.3. Change in annual mean water deficit projections using different $ET_0$ methods
Figure 8 shows the change in annual mean water deficit over all 9 GCMs for the RCP 4.5
trajectory in future period 1 (2030-2060) predicted by the ten different $ET_0$ methods used in this
study (a: Hargreaves, b: Blaney-Criddle, c: Hamon, d: Kharrufa, e: Irmak-Rn, f: Irmak-Rs, g:
Dalton, h: Meyer, i: Penman-Monteith, j: Priestley-Taylor). This figure clearly shows that the
changes in water deficit for future period 1 are diverse and depend strongly on the choice of $ET_0$
method. Except for the Hargreaves method (Fig. 8a) the temperature based methods (e.g.
Blaney-Criddle (Fig. 8b), Hamon (Fig. 8c) and Kharrufa (Fig. 8d)) predict drier conditions over
the continental US than the other methods. The mass transfer based methods (e.g Dalton (Fig.
8g) and Meyer (Fig. 8h)) predict generally wetter conditions over most of the continental US
compared to other methods. The combination method (Penman Monteith (Fig. 8i)), and the
radiation based methods (Irmak-Rn (Fig 8e), Irmak-Rs (Fig. 8f) and Priestley Taylor (Fig. 8j))
generally fall between the mass transfer based and temperature based methods, with the
combination methods producing slightly drier conditions. Although most methods predict similar
spatial patterns of water deficit over the continental US (generally drier conditions in the West,
Southwest and South and generally wetter elsewhere), the Hamon method predicts a different
pattern of water deficit over the Southwest, South and Northern Rockies and Plains regions.
3.4. Monte Carlo filtering
Monte Carlo filtering (Saltelli et al., 2008) was conducted to further investigate whether
projected wetter or drier future conditions (i.e. larger or smaller annual mean water deficit) could
be attributed to specific GCMs, $ET_0$ estimation methods, or RCP trajectories. Figures 9 shows
the histograms for wet conditions and dry conditions in future 2 over the Southeast US by GCM,
$ET_0$ method and RCP trajectory for the example month of July. Figure 10 shows similar
histograms for the Northern Rockies and Plains, a region with differing behavior from the
Southeast US. Table 3 shows the P-value results for the $X^2$- test for all months in both futures for
the Southeast and Northern Rockies and Plains regions. P-values greater than 0.05 (shaded in
grey) indicate the two histograms are not significantly different from each other. Tables 4 – 6
show the fraction of time that a particular GCM (Table 4), $ET_0$ method (Table 5), or RCP
trajectory (Table 6) projected drier future conditions in each of the nine US climate regions for
each month, with fractions higher than 0.5 shaded in grey.

## 4. Discussion

Drier conditions in southern regions (Southeast, South, Southwest and West) and wetter

conditions in northern regions (Northeast, Ohio Valley, Upper Midwest, Northern Rockies and
Plains and Northwest) are consistent (Fig. 4) with those reported by McAfee (2013) who used 3
$ET_0$ methods (Hamon, Priestley-Taylor and Penman-Monteith) to estimate global changes in $ET_0$
over the entire globe. As found by Baker and Huang (2014) for both CMIP3 and CMIP5
projections, mean $ET_0$ is projected to be higher in future period 2 than in future period 1, and
mean precipitation projections are approximately equivalent in future period 1 and future period
2. Thus the projected mean changes in water deficit for future period 2 (red lines in Fig. 4) are
larger in magnitude than the projected changes for future period 1 (blue lines). In all regions, and
for both future periods, the one standard deviation error bars bracket zero mean change
indicating large uncertainty in the projections throughout the year.

The choice of GCM is generally more important than the choice of RCP trajectory for

projected changes in P (Fig. 5). This is consistent with results found by Gaetani and Mohino
(2013) and Knutti and Sedláček (2012) who showed significant differences in precipitation
predictions among CMIP5 models. It should be noted that these results do not indicate that the
choice of RCP trajectory does not affect the change in precipitation, only that the choice of RCP
trajectory is less influential than the choice of GCM. There are no consistent seasonal patterns of
the first-order sensitivity coefficients for either GCM or RCP trajectory in either future period.

However, during the spring months, the sensitivity of change in P to choice of RCP trajectory increases substantially in future 2 compared to future 1 in the Northeast, Ohio Valley, Upper Midwest, South, Southwest and West regions.

Higher sensitivity of mean change in $ET_0$ to the choice of $ET_0$ estimation method than the choice of GCM (Fig. 6) are consistent with those found by Kingston et al. (2009) who showed that projected increase in $ET_0$ varied by more than 100% between $ET_0$ methods, and Schwalm et al. (2013) who found the choice of $ET_0$ estimation method is sensitive and even more influential than the choice of GCM in predicting $ET_0$. However, neither of these studies looked at the influence of RCP trajectory on $ET_0$ projections, which increases in future period 2 over future period 1, causing a decrease in the sensitivity coefficient of both GCM and $ET_0$ method in future 2. Burke and Brown (2008) evaluated uncertainties in the projection of future drought using several drought indices. They found that there are large uncertainties in regional changes in drought and changes in drought are dependent on both index definition and GCM ensemble members. Similarly, our results for the projected change in water deficit vary by region, depend strongly on the choice of GCM and $ET_0$ estimation method, but are relatively less sensitive to RCP trajectory (Fig. 7). These findings are similar to results reported by Orlowsky and Seneviratne (2013) who found that the greenhouse gas emission scenario uncertainty is not as important as differences among GCMs or internal climate variability when predicting Standardized Precipitation Index (SPI) and soil moisture (SMA). However, they also found that uncertainty due to greenhouse gas emission scenario increased in later future periods. Taylor et al. (2013) showed the patterns of changes in future drought were similar between the A1B scenario in CMIP3 and the RCP2.6 trajectory in CMIP5, reinforcing our finding that the choice of RCP trajectory is less important than the choice of GCM and $ET_0$ estimation method when estimating future water deficit.

Similar to the results of Kay and Davies (2008) and Bae et al. (2011) the results of our GSA show that the choice of $ET_0$ method has important implications when making future $ET_0$ projections and future water deficit projections (Fig. 8). Kingston et al. (2009) recommended the use of different $ET_0$ equations to evaluate global $ET_0$, and Wang et al. (2015) found that although different methods predict similar future $ET_0$, there are important differences in uncertainties due to $ET_0$ estimation methods and input data reliability. Currently many hydrological models use a

single evapotranspiration method for simulation, which may substantially increase the
uncertainty and reduce the reliability of future projections. Our results strongly indicate that an
ensemble of $ET_0$ estimation methods should be used to understand potential future water
availability and water deficit due to climate change.
Monte Carlo filtering results (Fig. 9 and 10, Table 3) indicate that GCM and $ET_0$ methods
both produce statistically significant different wet condition and dry condition histograms in both
the Southeast and Northern Rockies and Plains regions for almost all months in both future
periods. This indicates that particular GCMs and $ET_0$ methods tend to systematically produce
wet or dry conditions. Some GCMs (i.e. MIROC_ESM and BCC-CSM (Table 4)) and $ET_0$
methods (i.e. Priestley-Taylor, Blaney-Criddle, and Kharrufa (Table 5)) predict dry conditions a
majority of the time for all regions in both future periods. However, the remaining GCMs and
$ET_0$ methods project both wetter or drier futures depending on the region and future period.
Results in Tables 4 through 6 show that for the South, West and Southwest regions drier
conditions are predicted a majority of the time in both future periods by all GCMs and RCP
trajectories, and all $ET_0$ methods except Hargreaves. For RCP trajectory, P-values indicate the
histograms are statistically significantly different in fewer cases than for either GCM or $ET_0$
method for both future 1 and 2 (Table 3). These results are consistent with the first order
sensitivity coefficients results that showed the RCP trajectory is not as important a factor as
GCM or $ET_0$ method in driving differences in future projections, but that the sensitivity to choice
of RCP trajectory increases in future period 2.
GCMs estimate some climate variables, such as temperature, with higher confidence than
other variables (Randall et al., 2007). However, for some evapotranspiration estimation methods
the effect of temperature on evaporation is smaller than other climate variables (Linacre, 1994;
Roderick et al., 2009a, 2009b; Thom et al., 1981). We found that temperature and net radiation
from the CMIP5 GCMs show increasing trends over the 2005-2100 time period, while wind
speed and surface pressure are relatively constant (Fig. S-3). Because we considered various $ET_0$
estimation methods our results include the impacts of the different physics represented in the $ET_0$
methods, the projected changes each of the climate variables contributing to the different $ET_0$
methods, and the reliability of the predictions of each variable.

## 5. Summary and Conclusions

Future changes in precipitation and evapotranspiration will lead to changes in the hydrologic balance. This study clearly shows that the uncertainty caused by different GCMs, $ET_0$ methods, and RCP trajectories make actionable water resources planning based on climate change projections difficult. Understanding and quantifying how these projected changes vary with choice of GCM, $ET_0$ method and RCP trajectory is important for designing robust ensembles of scenarios to include in future water resources planning. This study assessed the future mean change in monthly precipitation, evapotranspiration and water deficit (P- $ET_0$) projected by CMIP5 simulations over the continental US and analyzed the sensitivity of the projected changes to the choice of GCM, $ET_0$ estimation method, and RCP trajectory. Nine GCMs, ten $ET_0$ estimation methods, and three RCP trajectories were included in the analyses. Variance-based global sensitivity analysis (Saltelli et al., 2010) was conducted in order to determine the relative contributions of the choice of GCMs, $ET_0$ estimation methods, and RCP trajectory to uncertainty in future prediction. Monte Carlo filtering was used to investigate whether particular GCMs, $ET_0$ methods, and/or RCP scenarios consistently led to wet or dry future projections.

The global sensitivity analyses showed that projected changes in precipitation are more sensitive to the choice of GCM than the choice of RCP trajectory over the entire continental US for both future periods. However, the choice of RCP trajectory becomes more important in future period 2. The most sensitive factor for the future $ET_0$ projections is the choice of $ET_0$ estimation method for all regions in both future periods. The first order sensitivity of projected change in future $ET_0$ to choice of RCP trajectory increases in future period 2 compared to future 1, with a concomitant decrease in the first order sensitivity to the choice of GCM. For projected change in future water deficit the choice of $ET_0$ method constitutes the dominant source of uncertainty in warmer months (May through September) and the choice of GCM is the dominant source of uncertainty in the cooler months (November through March) over all regions except the Southeast where the sensitivity of GCM and $ET_0$ method are roughly equal throughout the year. Sensitivity of change in future water deficit to RCP trajectory is very small for future period 1, but increased in future period 2.

Monte Carlo filtering results indicated that both GCMs and $ET_0$ methods produced
statistically different histograms for wetter or drier future conditions (i.e. larger or smaller mean
future water deficit) for almost all months in both future periods. Two GCMs (MIROC_ESM
and BCC-CSM) and three $ET_0$ methods (Priestley-Taylor, Blaney-Criddle, and Kharrufa)
predicted dry conditions a majority of the time for all regions in both future periods; however,
the remaining GCMs and $ET_0$ methods projected both wetter and drier futures depending on the
region.
Results of this study indicate that when predicting the effects of future climate on water
resources the choice of evapotranspiration method should be carefully evaluated. Rather than the
typical practice of using a single $ET_0$ method to drive a hydrologic model with future climate
projections, an ensemble of $ET_0$ methods should be used in addition to an ensemble of GCMs
and a variety of RCP trajectories. The GSA methodology adopted here assumed that all the
GCMs, $ET_0$ methods and RCP trajectories used in this study were equally appropriate for use in
all US regions (i.e. the sensitivity coefficients were evaluated by equally weighting each GCMs,
$ET_0$ method and RCP trajectory) which is likely not to be the case. When making future
projections  potential climate change on water resources Reliability Ensemble Averaging (REA)
(Giorgi and Mearns, 2002) or Bayesian-based indicator-weighting (Asefa and Adams, 2013;
Tebaldi et al., 2005; Xing et al., 2014) could be used to weight the results of an ensemble of
GCMs and ET methods based on how close the retrospective GCM- $ET_0$ method predictions
agree with past observations (bias criterion) and how well the future GCM- $ET_0$ -RCP
projections agree with other future GCM- $ET_0$ -RCP predictions (convergence criterion).
This study assumed that $ET_0$ methods that have been developed and parameterized based
on vegetation response to current $CO_2$ levels and climatic conditions will be valid under future
$CO_2$ levels and climatic conditions. Future research should explore the validity of this
assumption by incorporating potential changes in plant transpiration (e.g. stomatal conductance)
to changing $CO_2$ levels into the $ET_0$ estimation methodologies.

**Acknowledgements**

This research was supported by Tampa Bay Water and the University of Florida Water

Institute. We acknowledge the modeling groups participating in the Program for Climate Model
Diagnosis and Inter-comparison (PCMDI) for their role in making the CMIP5 (Coupled Model
Intercomparison Project) multi-model data set available.

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

Table 1. Description of reference evapotranspiration estimation methods used in this study ($ET_0$:
Reference evapotranspiration).

| Methods | Equations[1] | Reference |
|---|---|---|
| (a) Hargreaves | $ET_0 = 0.0135K_T S_0(T + 17.8)\sqrt{\delta_T}$ | Hargreaves and Allen (2003) |
| (b) Blaney-Criddle | $ET_0 = p(0.46T + 8.13)$ | Xu and Singh (2002) |
| (c) Hamon | $ET_0 = 0.55\delta_T^2 P_t$ | Xu and Singh (2002) |
| (d) Kharrufa | $ET_0 = 0.34pT^{1.3}$ | Xu and Singh (2002) |
| (e) Irmak-Rn | $ET_0 = 0.486 + 0.289R_n + 0.023T$ | Irmak et al. (2003) |
| (f) Irmak-Rs | $ET_0 = -0.611 + 0.149R_s + 0.079T$ | Irmak et al. (2003) |
| (g) Dalton | $ET_0 = (0.3648 + 0.07223u)(e_s - e_a)$ | Tabari et al. (2013) |
| (h) Meyer | $ET_0 = (0.375 + 0.05026u)(e_s - e_a)$ | Tabari et al. (2013) |
| (i) Penman-Monteith | $ET_0 = \dfrac{0.408\Delta(R_n - G) + \gamma \dfrac{900}{T + 273}u_2(e_s - e_a)}{\Delta + \gamma(1 + 0.34u_2)}$ | Allen et al. (1998) |
| (j) Priestley-Taylor | $ET_0 = \alpha \dfrac{\Delta}{\Delta + \gamma}\dfrac{(R_n - G)}{\lambda}$ | Allen et al. (1998) |

[1]Variables (estimated from CMIP5 outputs): G: Soil heat flux (assumed 0); $\gamma$: Psychrometric constant; T: Average
temperature; $u_2$: Wind speed at 2m surface; $e_s$: Saturated vapor pressure; $e_a$: Actual vapor pressure; $\Delta$: Slope vapor
pressure; $K_T$: Hargreaves-Samani coefficient; $S_0$: Extraterrestrial radiation (estimated by Julian date); $\delta_T$: Difference
between maximum and minimum temperature, p: Percentage of total daytime hours (Estimated by Julian date); $R_n$:
Net radiation; $R_s$: Solar radiation; $P_t$: Saturated water vapor density; u: Wind speed
Table 2. Description of the CMIP5 models used in this study.

| Model | Institute (country) | Resolutions | Calendar | Reference |
|---|---|---|---|---|
| (1) BNU-ESM | College of Global Change and Earth System Science, Beijing Normal University (China) | 2.8° lat × 2.8° lon | No leap | Ji et al. (2014) |
| (2) CSIRO-MK3-6-0 | University of New South Wales (Australia) | 1.87° lat × 1.87° lon | No leap | Rotstayn et al. (2012) |
| (3) GFDL-CM3 | NOAA/Geophysical Fluid Dynamics Laboratory (USA) | 2.0° lat × 2.5° lon | No leap | Guo et al. (2014) |
| (4) GFDL-ESM2G | NOAA/Geophysical Fluid Dynamics Laboratory (USA) | 2.0° lat × 2.5° lon | No leap | Taylor et al. (2012) |
| (5) MIROC-ESM | Atmosphere and Ocean Research Institute, National Institute for Environmental Studies, and Japan Agency for Marine-Earth Science and Technology (Japan) | 2.8° lat × 2.8° lon | Leap year | Watanabe et al. (2011) |
| (6) MPI-ESM-LR | Max Planck Institute for Meteorology (Germany) | 1.87° lat × 1.87° lon | Leap year | Block and Mauritsen (2013) |
| (7) MRI-CGCM3 | Meteorological Research Institute (Japan) | 1.12° lat × 1.12° lon | Leap year | Yukimoto et al. (2012) |
| (8) NorESM1-M | Norwegian Climate Centre (Norway) | 1.9° lat × 2.5° lon | No leap | Bentsen et al. (2013) |
| (9) BCC-CSM1.1 | Beijing Climate Center (China) | 2.8° lat × 2.8° lon | No leap | Xiao-Ge et al. (2013) |


Table 3. P-values of Chi-square two sample test for difference among wet condition versus dry condition pdfs Southeast U.S (SE US) and Northern Rockies and Plains (NRP; West North Central) U.S. (Shaded cells indicate pdfs are not statistically significantly different at p=0.05)

| Month | | Future 1 | | | Future 2 | | |
|---|---|---|---|---|---|---|---|
| | | GCM | $ET_0$ | RCP | GCM | $ET_0$ | RCP |
| SE US | 1 | 0.0000 | 0.0689 | 0.3701 | 0.0000 | 0.1823 | 0.1853 |
| | 2 | 0.0000 | 0.0889 | 0.4434 | 0.0000 | 0.0269 | 0.0000 |
| | 3 | 0.0000 | 0.0365 | 0.0306 | 0.0000 | 0.0000 | 0.1339 |
| | 4 | 0.0000 | 0.0000 | 0.6602 | 0.0000 | 0.0000 | 0.0001 |
| | 5 | 0.0000 | 0.0000 | 0.3223 | 0.0000 | 0.0000 | 0.0041 |
| | 6 | 0.0000 | 0.0000 | 0.0809 | 0.0000 | 0.0000 | 0.0006 |
| | 7 | 0.0000 | 0.0000 | 0.2855 | 0.0000 | 0.0000 | 0.0749 |
| | 8 | 0.0000 | 0.0000 | 0.2805 | 0.0000 | 0.0000 | 0.0074 |
| | 9 | 0.0000 | 0.0000 | 0.8646 | 0.0000 | 0.0000 | 0.0044 |
| | 10 | 0.0000 | 0.0000 | 0.0000 | 0.0000 | 0.0000 | 0.0001 |
| | 11 | 0.0000 | 0.0001 | 0.0000 | 0.0000 | 0.0001 | 0.2003 |
| | 12 | 0.0000 | 0.0117 | 0.3083 | 0.0000 | 0.0000 | 0.0000 |
| NRP | 1 | 0.0000 | 0.0000 | 0.1931 | 0.0000 | 0.0000 | 0.0000 |
| | 2 | 0.0000 | 0.0000 | 0.0010 | 0.0000 | 0.0000 | 0.7617 |
| | 3 | 0.0000 | 0.0000 | 0.0538 | 0.0000 | 0.0000 | 0.0769 |
| | 4 | 0.0000 | 0.0000 | 0.7882 | 0.0002 | 0.0000 | 0.8925 |
| | 5 | 0.0000 | 0.0000 | 0.4047 | 0.0000 | 0.0000 | 0.1103 |
| | 6 | 0.0000 | 0.0000 | 0.3839 | 0.0000 | 0.0000 | 0.0000 |
| | 7 | 0.0000 | 0.0000 | 0.5321 | 0.0001 | 0.0008 | 0.0000 |
| | 8 | 0.0000 | 0.0001 | 0.1544 | 0.0000 | 0.0686 | 0.0000 |
| | 9 | 0.0000 | 0.0000 | 0.4242 | 0.0000 | 0.0000 | 0.2002 |
| | 10 | 0.0000 | 0.0000 | 0.6688 | 0.0000 | 0.0213 | 0.0001 |
| | 11 | 0.0000 | 0.0000 | 0.1334 | 0.0000 | 0.0000 | 0.1948 |
| | 12 | 0.0000 | 0.0000 | 0.7617 | 0.0000 | 0.0000 | 0.6561 |

Table 4. The fraction of future dry conditions over all months by GCM (Future period 1 and 2).

| | GCM | SE | South | West | NR | NE | NW | UM | SW | Ohio |
|---|---|---|---|---|---|---|---|---|---|---|
| Future period 1 - Dry condition | BNU_ESM | 0.575 | 0.589 | 0.511 | 0.367 | 0.436 | 0.322 | 0.467 | 0.453 | 0.492 |
| | CSIRO_mk3_6_0 | 0.489 | 0.689 | 0.639 | 0.547 | 0.297 | 0.519 | 0.381 | 0.653 | 0.481 |
| | GFDL_CM3 | 0.414 | 0.608 | 0.686 | 0.419 | 0.403 | 0.525 | 0.383 | 0.647 | 0.425 |
| | GFDL_ESM2G | 0.731 | 0.900 | 0.758 | 0.453 | 0.486 | 0.486 | 0.397 | 0.828 | 0.617 |
| | MIROC_ESM | 0.631 | 0.594 | 0.822 | 0.625 | 0.636 | 0.708 | 0.686 | 0.658 | 0.611 |
| | MPI_ESM_LR | 0.375 | 0.747 | 0.694 | 0.542 | 0.597 | 0.611 | 0.558 | 0.756 | 0.575 |
| | MRI_CGCM3 | 0.494 | 0.592 | 0.639 | 0.400 | 0.544 | 0.553 | 0.350 | 0.547 | 0.506 |
| | NorESM1_M | 0.492 | 0.764 | 0.778 | 0.475 | 0.400 | 0.611 | 0.475 | 0.753 | 0.508 |
| | BCC_CSM | 0.728 | 0.739 | 0.828 | 0.642 | 0.603 | 0.614 | 0.564 | 0.822 | 0.656 |
| Future period 2 - Dry condition | BNU_ESM | 0.608 | 0.775 | 0.597 | 0.400 | 0.522 | 0.461 | 0.478 | 0.522 | 0.572 |
| | CSIRO_mk3_6_0 | 0.367 | 0.667 | 0.583 | 0.528 | 0.225 | 0.528 | 0.433 | 0.633 | 0.461 |
| | GFDL_CM3 | 0.467 | 0.767 | 0.789 | 0.461 | 0.514 | 0.542 | 0.508 | 0.794 | 0.469 |
| | GFDL_ESM2G | 0.722 | 0.831 | 0.694 | 0.478 | 0.519 | 0.525 | 0.397 | 0.672 | 0.581 |
| | MIROC_ESM | 0.672 | 0.686 | 0.897 | 0.742 | 0.731 | 0.728 | 0.700 | 0.739 | 0.664 |
| | MPI_ESM_LR | 0.442 | 0.800 | 0.778 | 0.519 | 0.542 | 0.639 | 0.450 | 0.800 | 0.450 |
| | MRI_CGCM3 | 0.508 | 0.703 | 0.581 | 0.422 | 0.481 | 0.528 | 0.439 | 0.517 | 0.556 |
| | NorESM1_M | 0.594 | 0.808 | 0.722 | 0.500 | 0.461 | 0.550 | 0.481 | 0.731 | 0.594 |
| | BCC_CSM | 0.628 | 0.697 | 0.875 | 0.708 | 0.567 | 0.708 | 0.556 | 0.825 | 0.603 |



Table 5. The fraction of future dry condition over all months by $ET_0$ estimation method and
region (Future period 1 and 2).

| | $ET_0$ | SE | South | West | NR | NE | NW | UM | SW | Ohio |
|---|---|---|---|---|---|---|---|---|---|---|
| Future period 1 -Dry condition | Hargreaves | 0.302 | 0.426 | 0.559 | 0.333 | 0.309 | 0.466 | 0.321 | 0.485 | 0.324 |
| | Blaney_Criddle | 0.738 | 0.880 | 0.898 | 0.840 | 0.738 | 0.762 | 0.784 | 0.904 | 0.769 |
| | Hamon | 0.633 | 0.818 | 0.667 | 0.531 | 0.494 | 0.497 | 0.457 | 0.713 | 0.549 |
| | Kharrufa | 0.883 | 0.957 | 0.889 | 0.636 | 0.667 | 0.698 | 0.636 | 0.886 | 0.738 |
| | Irmak_Rn | 0.522 | 0.673 | 0.694 | 0.491 | 0.512 | 0.556 | 0.494 | 0.679 | 0.580 |
| | Irmak_Rs | 0.525 | 0.722 | 0.731 | 0.463 | 0.485 | 0.546 | 0.460 | 0.679 | 0.556 |
| | Dalton | 0.364 | 0.503 | 0.583 | 0.340 | 0.343 | 0.426 | 0.296 | 0.509 | 0.380 |
| | Meyer | 0.367 | 0.531 | 0.596 | 0.346 | 0.324 | 0.435 | 0.290 | 0.512 | 0.367 |
| | PM | 0.534 | 0.685 | 0.694 | 0.472 | 0.469 | 0.525 | 0.481 | 0.676 | 0.540 |
| | PT | 0.608 | 0.719 | 0.750 | 0.515 | 0.552 | 0.590 | 0.515 | 0.753 | 0.608 |
| Future period 2 -Dry condition | Hargreaves | 0.352 | 0.506 | 0.605 | 0.420 | 0.355 | 0.491 | 0.380 | 0.537 | 0.361 |
| | Blaney_Criddle | 0.765 | 0.907 | 0.880 | 0.877 | 0.769 | 0.818 | 0.830 | 0.901 | 0.806 |
| | Hamon | 0.633 | 0.861 | 0.679 | 0.552 | 0.491 | 0.528 | 0.460 | 0.719 | 0.574 |
| | Kharrufa | 0.883 | 0.954 | 0.898 | 0.704 | 0.713 | 0.728 | 0.682 | 0.883 | 0.784 |
| | Irmak_Rn | 0.515 | 0.738 | 0.710 | 0.494 | 0.491 | 0.574 | 0.503 | 0.685 | 0.543 |
| | Irmak_Rs | 0.534 | 0.796 | 0.753 | 0.485 | 0.497 | 0.562 | 0.478 | 0.719 | 0.562 |
| | Dalton | 0.349 | 0.596 | 0.620 | 0.389 | 0.358 | 0.475 | 0.315 | 0.540 | 0.373 |
| | Meyer | 0.352 | 0.596 | 0.630 | 0.383 | 0.349 | 0.488 | 0.309 | 0.546 | 0.361 |
| | PM | 0.543 | 0.744 | 0.701 | 0.475 | 0.485 | 0.531 | 0.463 | 0.679 | 0.528 |
| | PT | 0.639 | 0.784 | 0.765 | 0.509 | 0.562 | 0.593 | 0.515 | 0.716 | 0.608 |


Table 6. The fraction of future dry condition over all months by RCP trajectory and region
(Future period 1 and 2).

|  | RCP | SE | South | West | NR | NE | NW | UM | SW | Ohio |
|---|---|---|---|---|---|---|---|---|---|---|
| Future period 1 -Dry condition | 2.6 | 0.551 | 0.657 | 0.665 | 0.507 | 0.502 | 0.543 | 0.495 | 0.644 | 0.553 |
| | 4.5 | 0.553 | 0.698 | 0.739 | 0.515 | 0.475 | 0.554 | 0.482 | 0.731 | 0.556 |
| | 8.5 | 0.539 | 0.719 | 0.715 | 0.468 | 0.491 | 0.554 | 0.443 | 0.665 | 0.515 |
| Future period 2 -Dry condition | 2.6 | 0.516 | 0.649 | 0.657 | 0.486 | 0.524 | 0.515 | 0.465 | 0.617 | 0.545 |
| | 4.5 | 0.490 | 0.731 | 0.712 | 0.510 | 0.476 | 0.584 | 0.494 | 0.658 | 0.528 |
| | 8.5 | 0.664 | 0.864 | 0.803 | 0.590 | 0.520 | 0.637 | 0.521 | 0.803 | 0.577 |


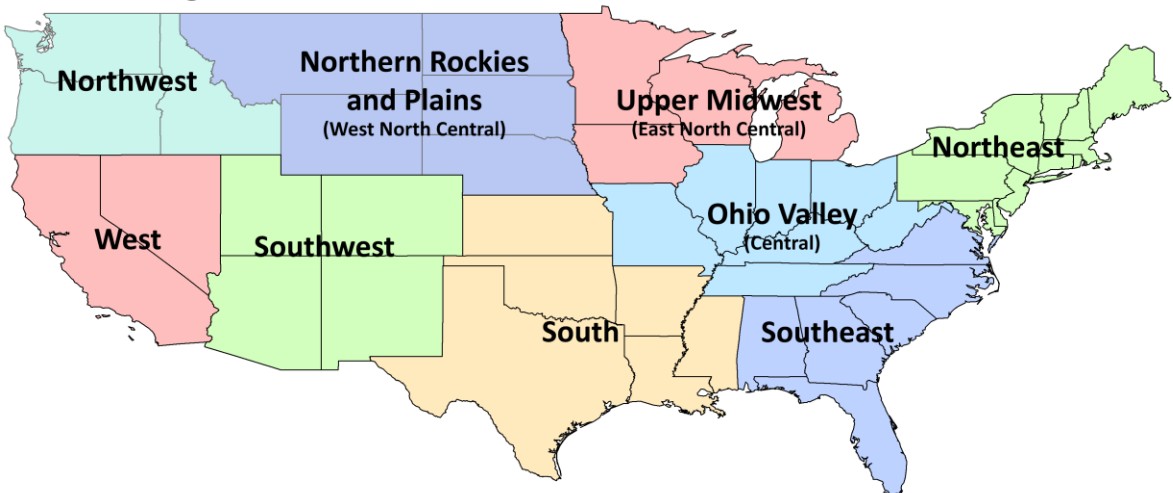


Figure 1. US climate regions identified by National Climate Data Center (Adapted from Karl and

Koss, 1984, https://www.ncdc.noaa.gov/monitoring-references/maps/us-climate-regions.php)

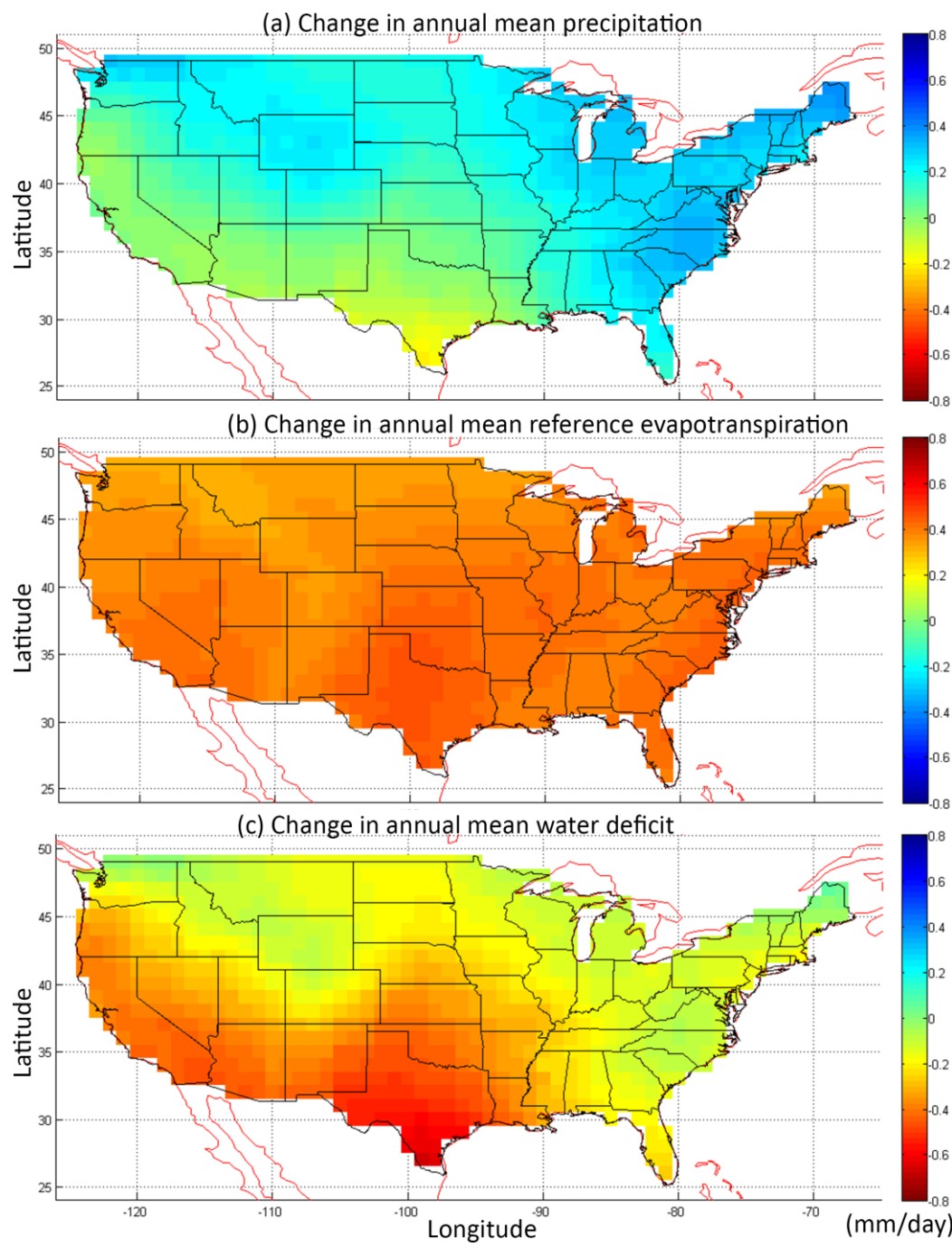

Figure 2. The change in the annual mean (a) P, (b) $ET_0$, and (c) $P - ET_0$ over U.S. All units are mm/day and the change is defined as the mean of 2070-2100 minus that of 1950-2005. These changes are averaged over all GCMs, $ET_0$ estimation methods, and RCP trajectories.

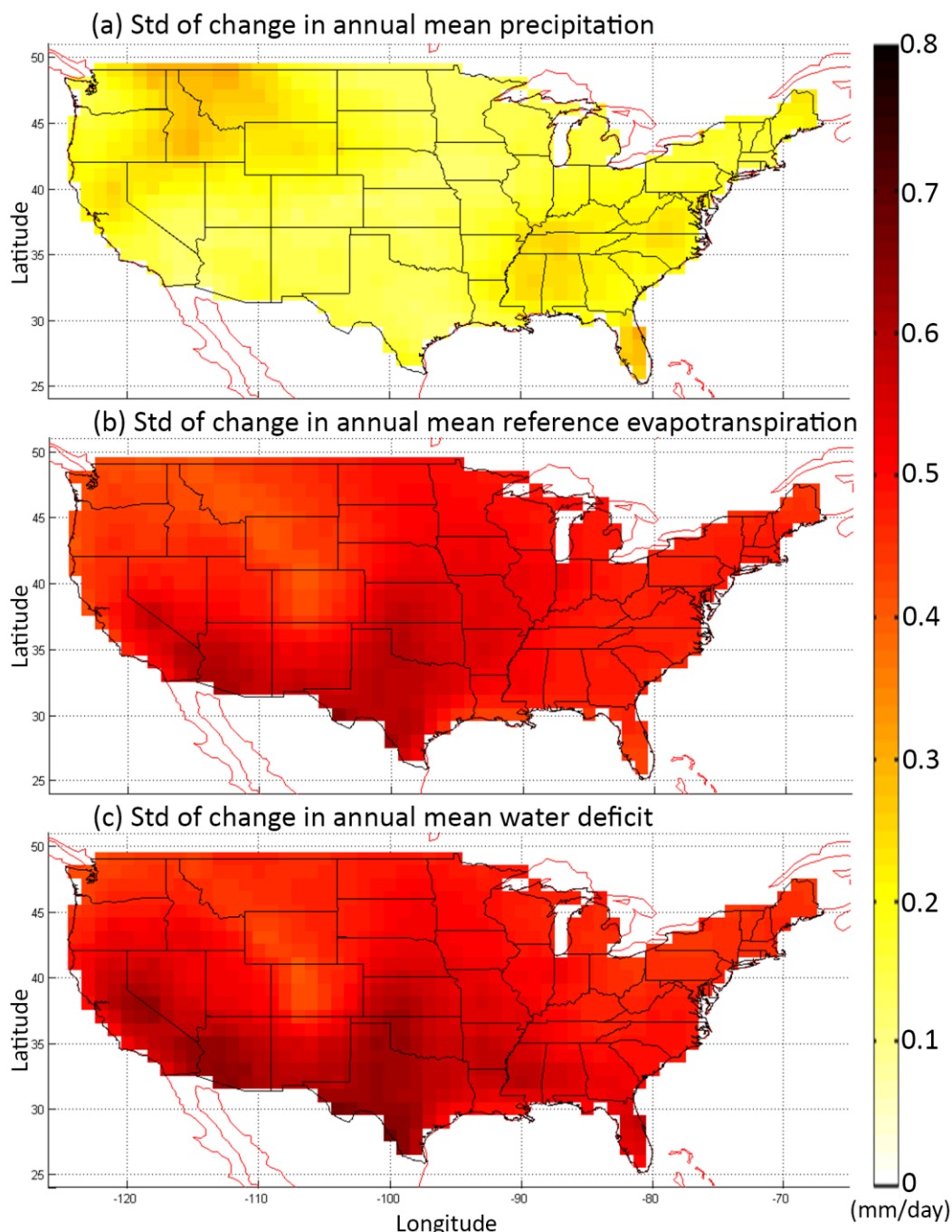

662

Figure 3. The standard deviation of the change in the annual mean (a) P, (b) $ET_0$, and (c) $P - ET_0$
over U.S. All units are mm/day and the change is defined as the average of 2070-2100 minus that
of 1950-2005. The standard deviations are estimated over all GCMs, $ET_0$ estimation methods,
and RCP trajectories.

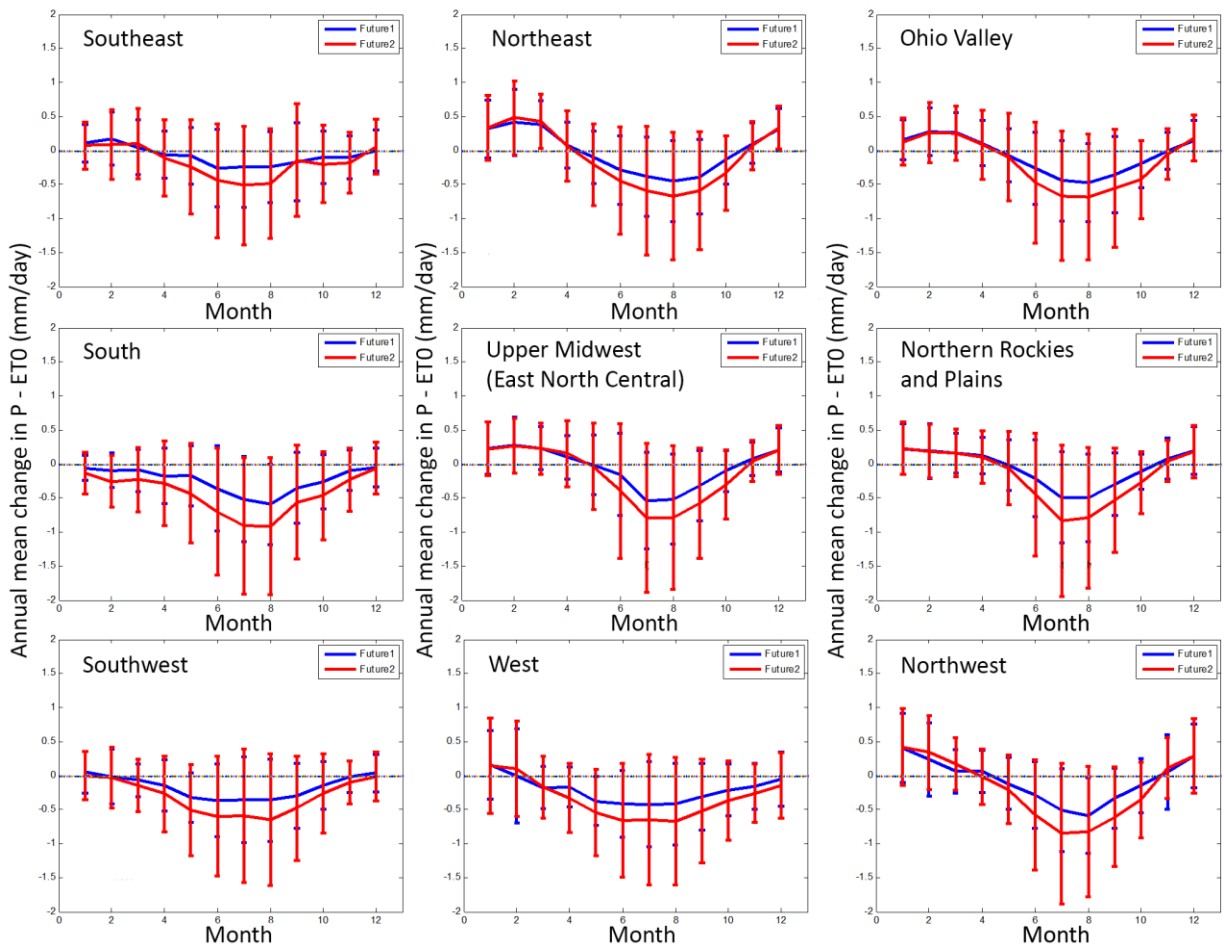

667

Figure 4. The change of monthly mean water deficit ($P - ET_0$) over 9 different regions. Blue

lines represent future 1 period (2030-2060), and red lines represent future 2 period (2070-2100).

Error bars represent one standard deviation of each values. The change is defined as the mean of

future periods minus that of retrospective period (1950-2005).

672

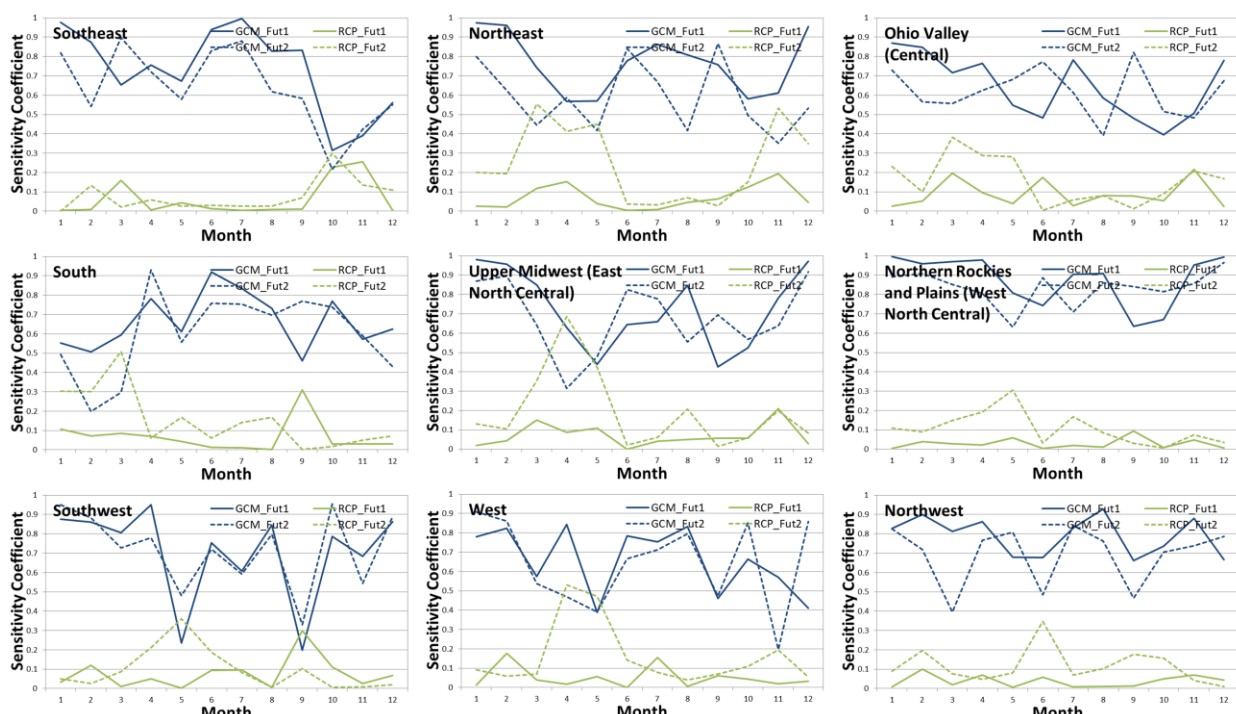

Figure 5. First order sensitivity analysis results of change in precipitation. Solid lines represent the future period 1 (2030-2060) and dotted lines represent the future period 2 (2070-2100). Blue lines represent the first order effect of GCMs and green lines represent the first order effect of RCPs.

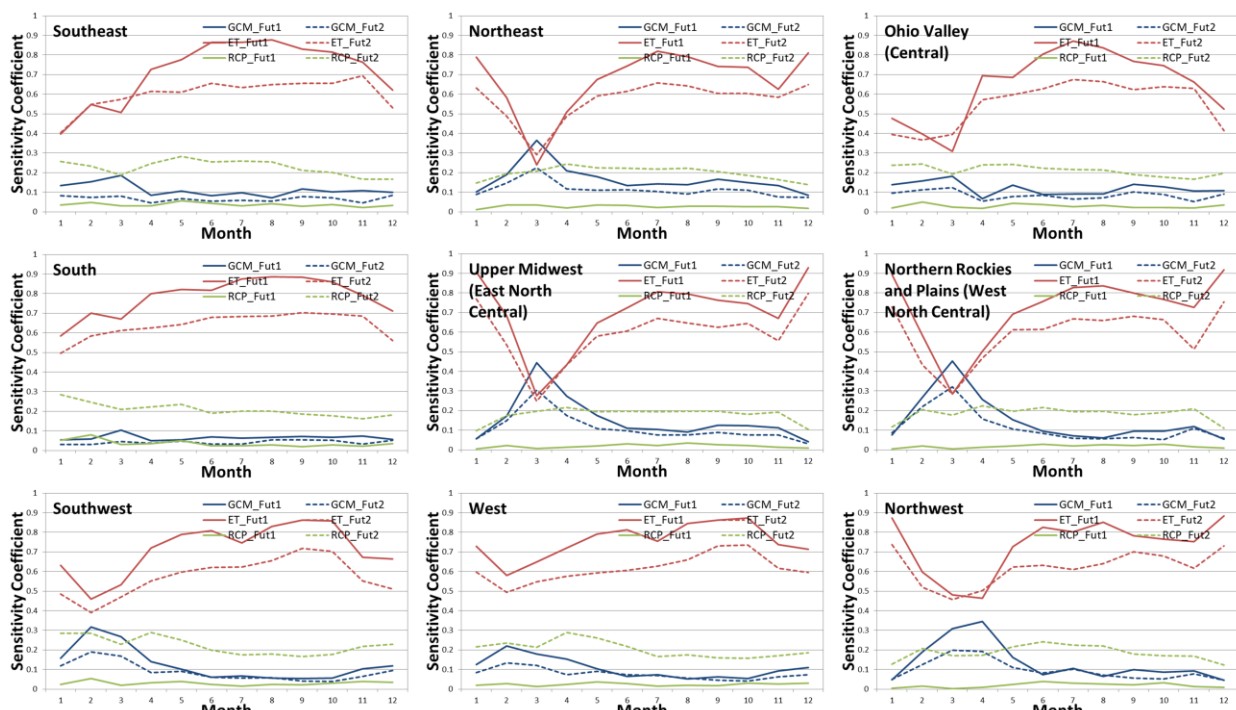


Figure 6. First order sensitivity analysis results of change in reference evapotranspiration. Solid
lines represent the future period 1 (2030-2060) and dotted lines represent the future period 2
(2070-2100). Blue lines represent the first order effect of GCMs, red lines represent the first
order effect of $ET_0$ estimation methods and green lines represent the first order effect of RCPs.

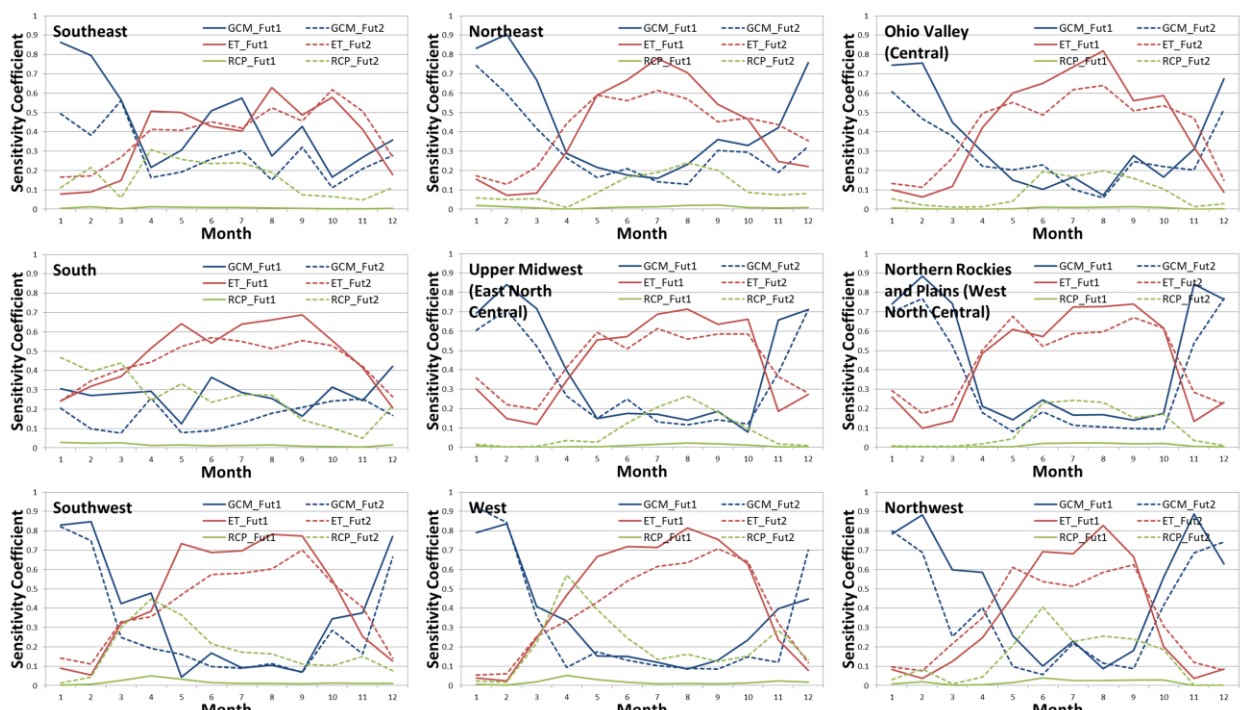


Figure 7. First order sensitivity analysis results of change in P - $ET_0$. Solid lines represent the

future period 1 (2030-2060) and dotted lines represent the future period 2 (2070-2100). Blue

lines  represent the first order effect of GCMs, red lines represent the first order effect of $ET_0$

estimation methods and green lines represent the first order effect of RCPs.

690

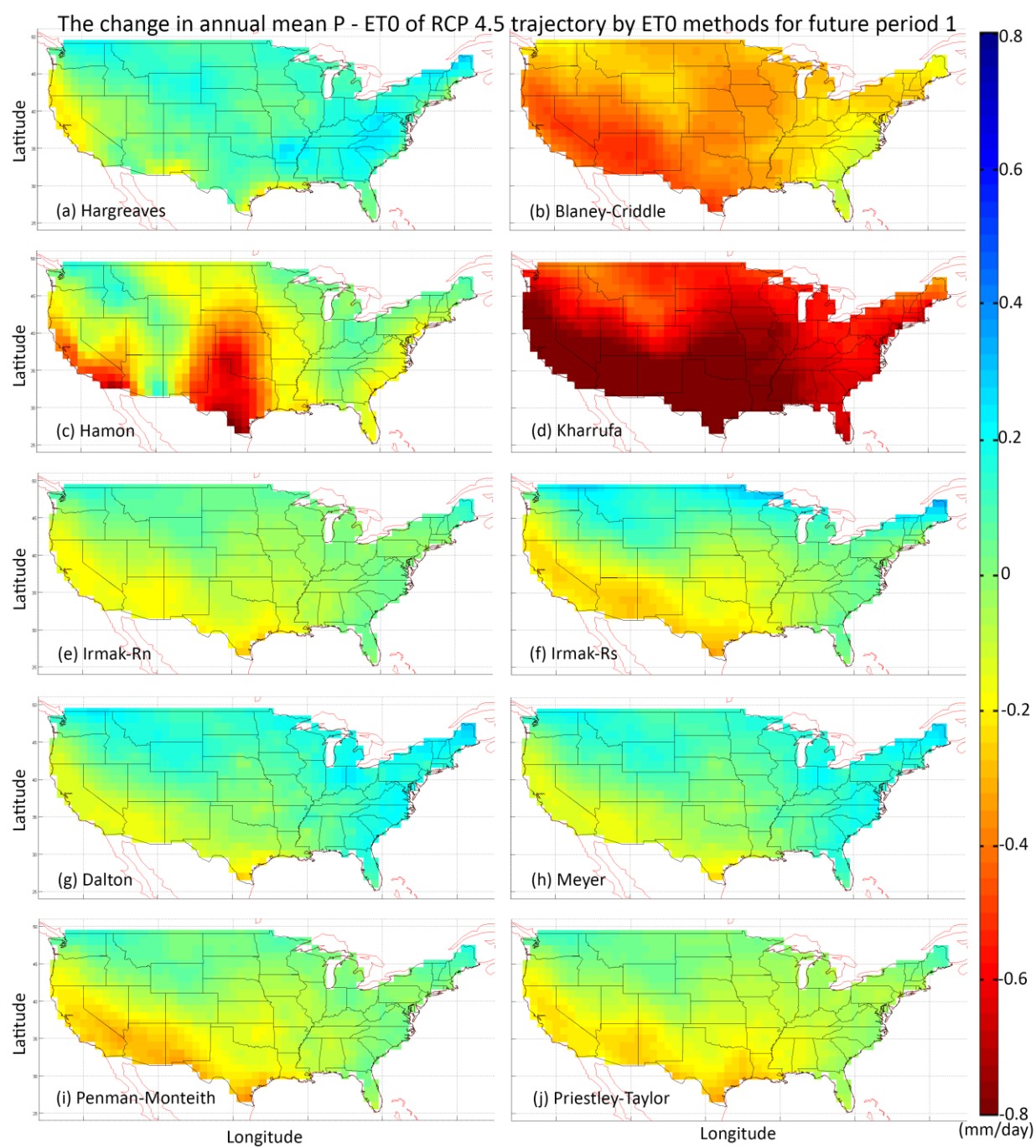

691

Figure 8. The change in the annual mean $P - ET_0$ of RCP 4.5 scenario by 10 different

evapotranspiration methods. All units are mm/day and the change is defined as the mean of

2030-2060 minus that of 1950-2005. (All results are interpolated to 1 degree * 1 degree grids and

averaged over 9 different GCMs)

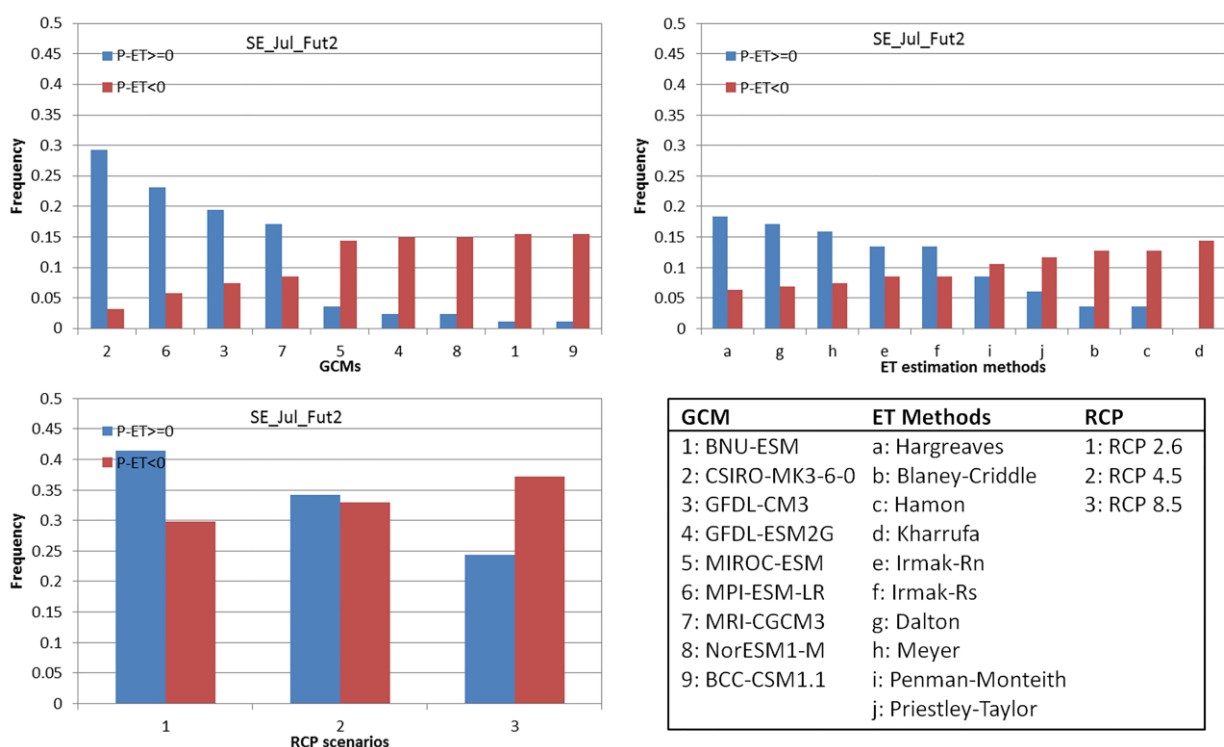


Figure 9. Histograms for projected future 2 wet conditions and dry conditions in the Southeast
US by GCM, $ET_0$ method and RCP trajectory for the month of July.

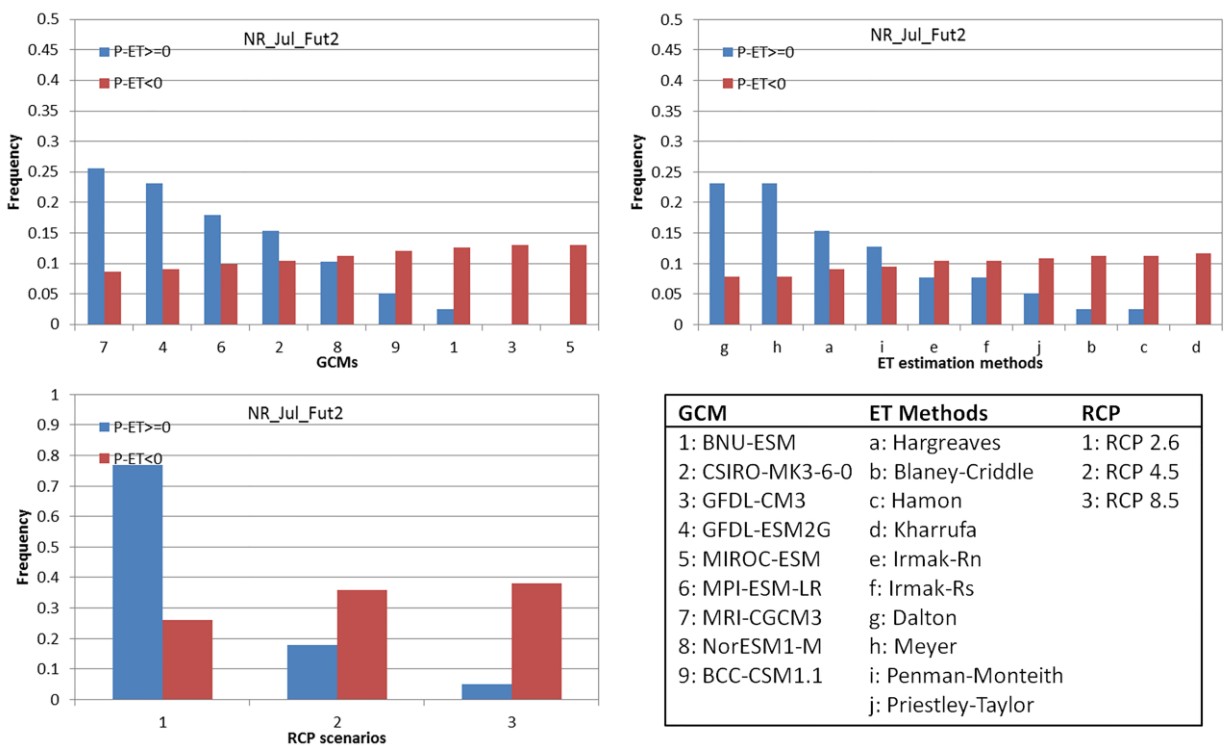


Figure 10. Histograms for projected future 2 wet conditions and dry conditions in the Northern
Rockies and Plains US by GCM, $ET_0$ method and RCP trajectory for the month of July.

**Appendix A: Supplemental figures**

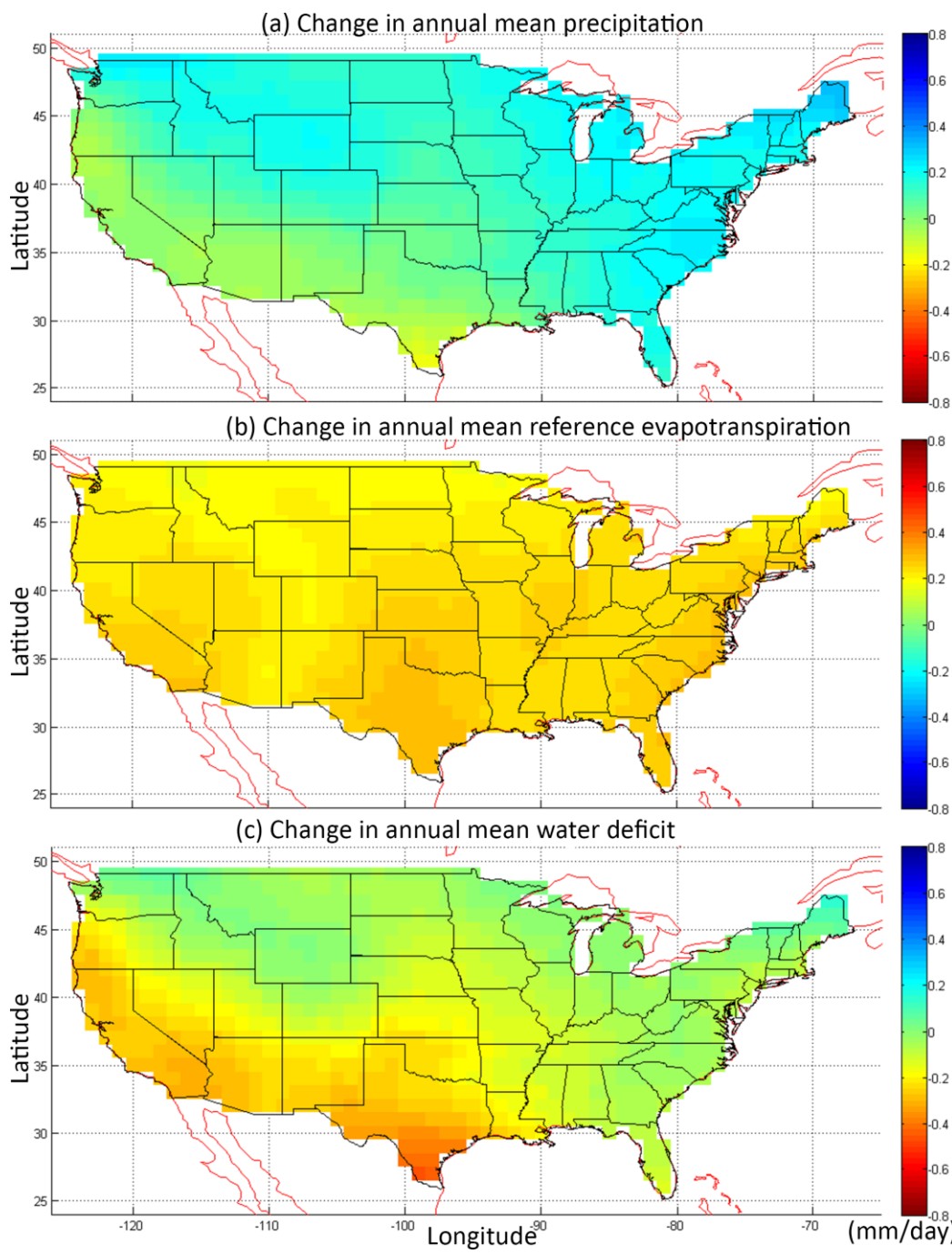

Fig. S-1 The change in the annual mean (a) P, (b) $ET_0$, and (c) $P - ET_0$ over U.S. All units are mm/day and the trend is defined as the average of 2030-2060 minus that of 1950-2005.

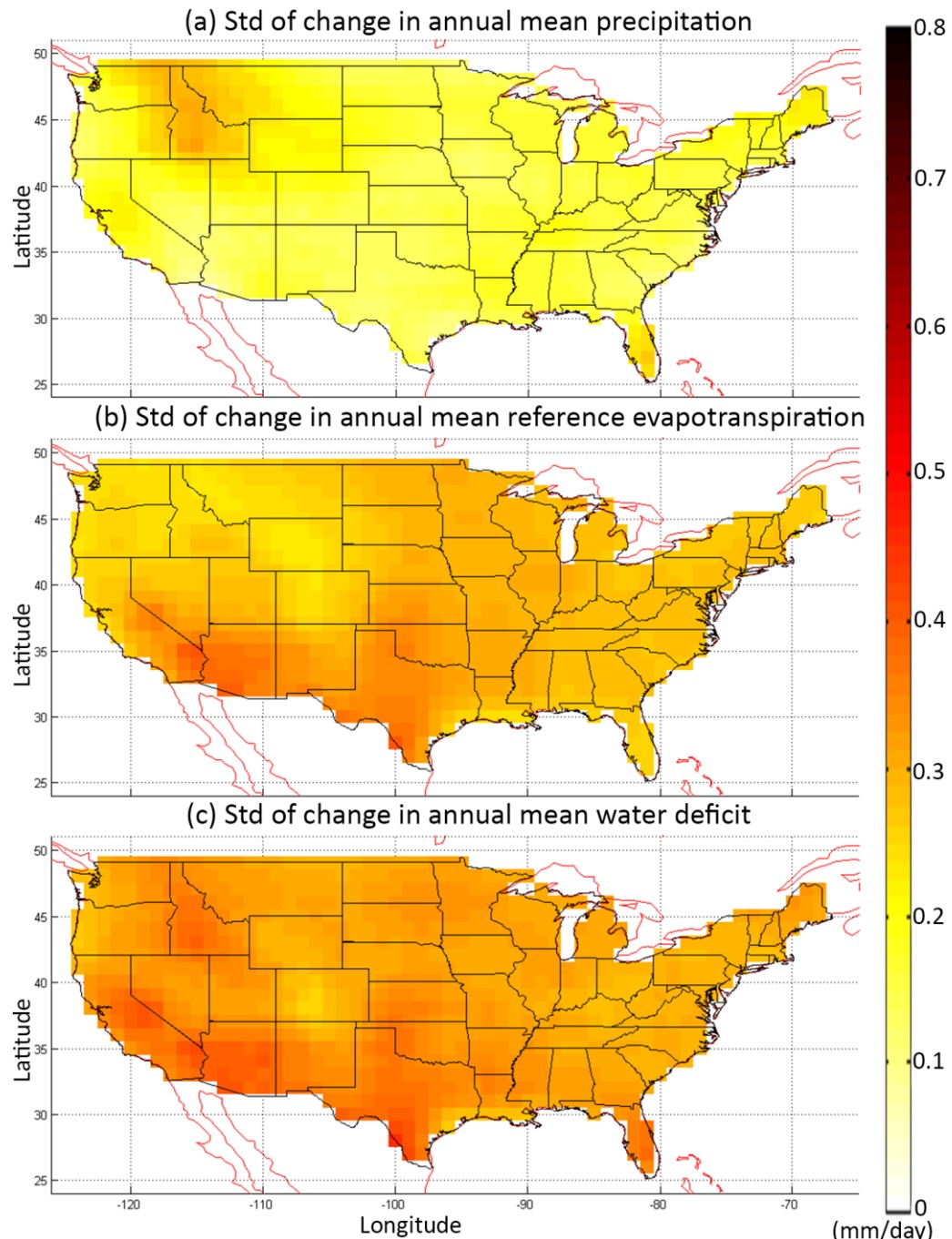


Fig. S-2 The standard deviation of the change in the annual mean (a) P, (b) $ET_0$, and (c) $P - ET_0$
over U.S. All units are mm/day and the trend is defined as the average of 2030-2060 minus that
of 1950-2005.


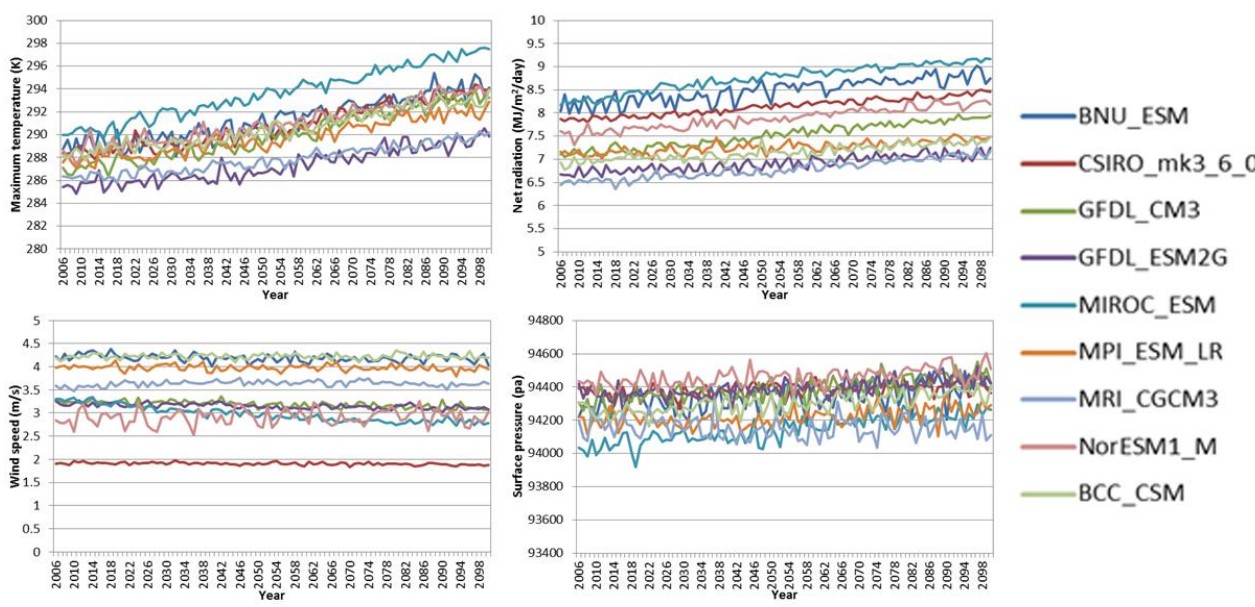

Fig. S-3 Mean maximum temperature, net radiation, wind speed at 2 m surface, and surface
pressure of CMIP5 for future period (RCP 8.5).