# Peer review of "Sensitivity of future Continental United States water deficit"

_Hydrology and Earth System Sciences, 2015_

## Referee Comment (RC1) · Anonymous Referee #1 · 26 Feb 2016

Summary

This is a well written and interesting paper that addresses a topical subject. Although there a few issues relating to structure & reference to related studies, I enjoyed reading this work and consider it generally suitable for publication in HESS, subject to the relatively minor issues mentioned below.

Major issues

1. Abstract, first sentence, and elsewhere. The authors need to clarify immediately that in this case, water availability refers to the meteorological water balance (i.e. P-PET).

[Figure]

Particularly in a hydrology-related journal such as HESS, water availability implies surface hydrological processes as well – in which case future water availability would depend on many other factors as well (e.g. irrigation abstractions, land use, water management strategies).

2. The Introduction section needs to better acknowledge that method-based PET uncertainty under climate change has been explored beyond just the meteorological water balance, to consider river flow as well (via hydrological models). Such studies include:

Bae, D.H., Jung, I.W. & Lettenmaier, D.P. 2011 Hydrologic uncertainties in climate change from IPCC AR4 GCM simulations of the Chungju Basin, Korea. Journal of Hydrology 401 90-105.

Kay, A.L. & Davies, H.N. 2008 Calculating potential evaporation from climate model data: A source of uncertainty for hydrological climate change impacts. Journal of Hydrology 358 221-239.

Koedyk, L.P. & Kingston, D.G. 2016, Potential evapotranspiration method influence on climate change impacts on river flow: a mid-latitude case study. Hydrology Research DOI: 10.2166/nh.2016.152.

Thompson, J.R., Green, A.J. & Kingston, D.G. 2014 Potential evapotranspiration-related uncertainty in climate change impacts on river flow: An assessment for the Mekong River basin. Journal of Hydrology 510 259-279.

3. The results and discussion are combined into a single section. Although I generally prefer these to be separated, the section is well written. At the very least, I would like to see the different aspects of the analysis divided into sub-sections, to help the reader follow the steps in the analysis.

4. P11, line 13: referring back to point 2 – yes, hydrological modelling studies that use only one PET method effectively ignore PET uncertainty, but there have been a series of studies that explicitly investigate this.

Minor/technical issues

5. According to the IPCC AR4 Glossary (http://www.ipcc.ch/pdf/assessment-report/ar5/wg1/WG1AR5_AnnexIV_FINAL.pdf), the acronym GCM stands for General Circulation Model. I suggest avoiding the term Global Climate Model and replacing with General Circulation Model.

6. P4, line 9: Priestley-Taylor is misspelt.

8. P5, line 27: Priestley-Taylor is a radiation based method – it only requires the slope of the vapour pressure curve (derived from temperature) and net radiation.

9. P6, line 3: RET is not defined in the paper. I presume RET means reference ET, but the commonly used abbreviation for this is ET0 (as used in the Table 1 caption).

10. P6. On line 3 precipitation is abbreviated to P; on line 5 it is abbreviated pr.

11. P7, line 11: spell out the number in this instance: nine, not 9 climate regions.

12. P10, line 15: typo: "sKingston".

13. P11, line 11: the acronym GSA is undefined.

---

## Referee Comment (RC2) · Anonymous Referee #2 · 12 Mar 2016

General comments The future projection of water availability is important for understanding the response of the hydrological regime to climate change and improvement to regional strategy for water resources management. However, projected availability is still a crucial challenge since uncertainty exists in estimated projected availability. Although the authors made an interesting investigation on the sensitivity of future water availability to GCM, RET estimation method and emission scenario, there are several aspects that need to be further considered and improved before it is considered for publication in HESS.

Specific comments (1) Before using the GCMs output to force hydrological model (even estimate RET), the some forms of prior bias correction are always conducted due that GCM often show strong bias over historic period (Wood et al., 2002; 2004). I can only believe the authors use the raw data causing I did not find any information associated with the bias correction description in the paper. So how about the matching degree between the GCM-simulated variables and historical observation? And whether some bias correction jobs should be done before employing these GCMs output. (2) GCM-simulated temperature is commonly considered to have high confidence than other climatic variables such as vapor pressure and radiation (Randall et al., 2007). The differences of estimated ET between temperature-based ET equations and radiation-based equations maybe due to the uncertain input data quality rather than the method selection as the authors declared. (3) In fact, temperature-based equations have been considered not competent in RET change (e.g., Roderick et al., 2009) due that a steady increase in temperature over time will translate into a calculated steady increase in evapotranspiration. Generally, using combination equations maybe more suitable for projection future RET. However, as the above comment pointed out, the GCM-simulated temperature was also widely considered to have relatively high confidence in comparison with other meteorological variables. The different combinations between methods and data should be discussed (see some literatures, Kingston et al., 2009; Wang et al., 2015). Some other minor comments (4) ET always mean actual evapotranspiration, it maybe better use RET/ET0 to represent reference evapotranspiration. (5) It is better to divide the results into several sub-sections. (6) Results should be presented as such and not mingled with explanations (analysis section), so please separate the results section and discussion section

Used literatures [1] Kingston, D.G., Todd, M.C., Taylor, R.G., Thompson, J.R., Arnell, N.W., 2009. Uncertainty in the estimation of potential evapotranspiration under climate change. Geophys. Res. Lett. 36, L20403. http://dx.doi.org/10.1029/2009GL040267. [2] Randall, D.A. et al., 2007. Climate models and their evaluation. In: Solomon, S.D. et al. (Eds.), Climate Change 2007: The Physical Science Basis. Contribution of Working

Group 1 to the Fourth Assessment Report of the Intergovernmental Panel on Climate Change. Cambridge Univ. Press, Cambridge, U.K., pp. 589–662. [3] Roderick, M.L., Hobbins, M.T., Farquhar, G.D., 2009. Pan evaporation trends and the terrestrial water balance. II. Energy balance and interpretation. Geogr. Campass 3, 761–780 [4] Wang W, Xing W., Shao Q., 2015. How large are uncertainties in future projection of reference evapotranspiration through different approaches? Journal of Hydrology, 524, 696-700 [5] Wood AW, Leung LR, Sridhar V, Lettenmaier DP (2004) Hydrological implications of dynamical and statistical approaches to downscaling climate model outputs. Climate Change 62:189–216. [6] Wood AW, Maure EP, Kumar A, Lettenmaier DP (2002) Long-range experimental hydrological forecasting for the eastern United States. J Geophys Res 107(D20):4429. doi:10.1029/2001JD00659

---

## Author Comment (AC1) · 13 Apr 2016

MS No.: HESS-2015-408R1; MS Type: Research article

We appreciate the thoughtful comments from the reviewers, which have helped us to improve the original manuscript. We explain in detail how we responded to each of the reviewer's comments, with line numbers referring to the revised manuscript unless otherwise noted. We changed our title to "Sensitivity of future Continental United States water deficit projections to General Circulation Model, evapotranspiration estimation method, and greenhouse gas emission scenario" in response to reviewers comment.

[Figure]

In addition, we upload revised manuscript and responses to reviewers as our supplemental material.

[1] Referee review: Abstract, first sentence, and elsewhere. The authors need to clarify immediately that in this case, water availability refers to the meteorological water balance (i.e. P-PET). Particularly in a hydrology-related journal such as HESS, water availability implies surface hydrological processes as well – in which case future water availability would depend on many other factors as well (e.g. irrigation abstractions, land use, water management strategies).

Author's response: We agree this could have been confusing. We replaced the term "water availability" by "water deficit" throughout the manuscript, and defined it early in the abstract and in body of the manuscript in order to clarify this.

[2] Referee review: The Introduction section needs to better acknowledge that method-based PET uncertainty under climate change has been explored beyond just the meteorological water balance, to consider river flow as well (via hydrological models). Such studies include: Bae, D.H., Jung, I.W. & Lettenmaier, D.P. 2011 Hydrologic uncertainties in climate change from IPCC AR4 GCM simulations of the Chungju Basin, Korea. Journal of Hydrology 401 90-105. Kay, A.L. & Davies, H.N. 2008 Calculating potential evaporation from climate model data: A source of uncertainty for hydrological climate change impacts. Journal of Hydrology 358 221-239. Koedyk, L.P. & Kingston, D.G. 2016, Potential evapotranspiration method influence on climate change impacts on river flow: a mid-latitude case study. Hydrology Research DOI: 10.2166/nh.2016.152. Thompson, J.R., Green, A.J. & Kingston, D.G. 2014 Potential evapotranspiration related uncertainty in climate change impacts on river flow: An assessment for the Mekong River basin. Journal of Hydrology 510 259-279.

Author's response: We introduced the references suggested in the introduction section and discussed differences among these studies and our study in the discussion section. For example after line 19 on page 4 we added: "Kay and Davies (2008) compared

the performance of the Penman-Monteith equation and a simple temperature-based evapotranspiration method using climate data from five global and eight regional climate models over Britain. They found that the two methods showed very different changes in potential evapotranspiration for the period 2071-2100 under the A2 emission scenario, and different flow predictions for three catchments when the data were used to force a rainfall-runoff model. Kay and Davies results suggest that hydrological prediction uncertainty due to potential evapotranspiration formulation was smaller than that due to GCM structure or RCM structure for their study region. Bae et al. (2011) evaluated the uncertainty contributed by choice of GCM and hydrologic model for the Chungju Dam basin, Korea. They found that hydrologic model structural differences contributed greater uncertainty than GCM selection to winter runoff prediction. Koedyk and Kingston (2016) found that for the Waikaia River, New Zealand potential evapotranspiration method contributed more uncertainty than GCM selection when predicting potential evapotranspiration, but that runoff predictions were more sensitive to GCMs than to potential evapotranspiration methods. Thompson et al. (2014) evaluated the effect of using different GCMs and different potential evapotranspiration methods on discharge predictions for the Mekong River in Southeast Asia and found that GCM-related uncertainty was greater than the potential evapotranspiration method related uncertainty. Our study adds to the literature by comprehensively evaluating the relative sensitivity of future P, ET0 and water deficit (defined here as P- ET0) projections to choice of GCM, ET0 method and RCP trajectory over the continental US."

[3] Referee review: The results and discussion are combined into a single section. Although I generally prefer these to be separated, the section is well written. At the very least, I would like to see the different aspects of the analysis divided into subsections, to help the reader follow the steps in the analysis.

Author's response: We divided the previously combined section into separate results and discussion sections as suggested.

[4] Referee review: P11, line 13: referring back to point 2 – yes, hydrological modelling

studies that use only one PET method effectively ignore PET uncertainty, but there have been a series of studies that explicitly investigate this.

Author's response: In addition to the revisions to the introductions noted in point 2 above, we changed the sentence on line 13, page 11 from "Many hydrological models use a single evapotranspiration method for simulation, which may substantially increase the uncertainty, and reduce the reliability of future projections." to "Similar to the results of Kay and Davies (2008) and Bae et al. (2011) the results of our GSA show that the choice of ET0 method has important implications when making future ET0 projections and future water deficit projections (Fig. 8). Kingston et al. (2009) recommended the use of different ET0 equations to evaluate global ET0, and Wang et al. (2015) found that although different methods predict similar future ET0, there are important differences in uncertainties due to ET0 estimation methods and input data reliability. Currently many hydrological models use a single evapotranspiration method for simulation, which may substantially increase the uncertainty and reduce the reliability of future projections. Our results strongly indicate that an ensemble of ET0 estimation methods should be used to understand potential future water availability and water deficit due to climate change."

[5] Referee review: According to the IPCC AR4 Glossary (http://www.ipcc.ch/pdf/assessmentreport/ar5/wg1/WG1AR5_AnnexIV_FINAL.pdf), the acronym GCM stands for General Circulation Model. I suggest avoiding the term Global Climate Model and replacing with General Circulation Model.

Author's response: We replaced 'Global Climate Model' with 'General Circulation Model' throughout the manuscript.

[6] Referee review: P4, line 9: Priestley-Taylor is misspelt.

Author's response: We replaced 'Preistly-Taylor' with 'Priestley-Taylor'.

[8; There's no 7th comment in the review note.] Referee review: P5, line 27: Priestley-

Taylor is a radiation based method – it only requires the slope of the vapour pressure curve (derived from temperature) and net radiation.

Author's response: We changed the classification of the Priestley-Taylor method to a radiation based method.

[9] Referee review: P6, line 3: RET is not defined in the paper. I presume RET means reference ET, but the commonly used abbreviation for this is ET0 (as used in the Table 1 caption).

Author's response: We have changed the abbreviation for reference ET to ET0 throughout the manuscript.

[10] Referee review: P6. On line 3 precipitation is abbreviated to P; on line 5 it is abbreviated pr.

Author's response: The paragraph on P.6 line 3 explains the CMIP5 archive. In the CMIP5 archive they use different abbreviations for precipitation and other climate variables than are conventionally used in hydrology and than we use in this manuscript. We have revised the paragraph to note these differences. "Variables directly used from the CMIP5 monthly model output included precipitation (pr, P in this study), maximum and minimum temperature (tasmax and tasmin), radiation (rlds, rlus, rsds, and rsus), air pressure (psl and ps), and wind speed (sfcWind). The abbreviations for these variables are as defined in the CMIP5 archive and explained in the PCMDI server (Program For Climate Model Diagnosis and Intercomparison, http://cmip-pcmdi.llnl.gov/cmip5/docs/standard_output.pdf)."

[11] Referee review: P7, line 11: spell out the number in this instance: nine, not 9 climate regions.

Author's response: We replaced '9' with 'nine'.

[12] Referee review: P10, line 15: typo: "sKingston".

Author's response: We replaced 'sKingston' with 'Kingston'.

[13] Referee review: P11, line 11: the acronym GSA is undefined.

Author's response: We defined GSA in the revised introduction section. "Global sensitivity analysis (GSA) apportions the total output uncertainty simultaneously onto all the uncertain input factors described by marginal probability density functions, and thus is preferred over local, one factor at a time, sensitivity analysis (Homma and Saltelli, 1996; Saltelli, 1999)."

[Additional Literature cited] Bae, D. H., Jung, I. W. and Lettenmaier, D. P.: Hydrologic uncertainties in climate change from IPCC AR4 GCM simulations of the Chungju Basin, Korea, J. Hydrol., 401(1-2), 90–105, doi:10.1016/j.jhydrol.2011.02.012, 2011. Homma, T. and Saltelli, A.: Importance measures in global sensitivity analysis of nonlinear models, Reliab. Eng. Syst. Saf., 52(1), 1–17, doi:10.1016/0951-8320(96)00002-6, 1996. Kay, A. L. and Davies, H. N.: Calculating potential evaporation from climate model data: A source of uncertainty for hydrological climate change impacts, J. Hydrol., 358(3-4), 221–239, doi:10.1016/j.jhydrol.2008.06.005, 2008. Kingston, D. G., Todd, M. C., Taylor, R. G., Thompson, J. R. and Arnell, N. W.: Uncertainty in the estimation of potential evapotranspiration under climate change, Geophys. Res. Lett., 36(20), L20403, doi:10.1029/2009GL040267, 2009. Koedyk, L. P. and Kingston, D. G.: Potential evapotranspiration method influence on climate change impacts on river flow: a mid-latitude case study, Hydrol. Res., doi:10.2166/nh.2016.152, 2016. Saltelli, A.: Sensitivity analysis: Could better methods be used?, J. Geophys. Res., 104(D3), 3789, doi:10.1029/1998JD100042, 1999. Thompson, J. R., Green, A. J. and Kingston, D. G.: Potential evapotranspiration-related uncertainty in climate change impacts on river flow: An assessment for the Mekong River basin, J. Hydrol., 510, 259–279, doi:10.1016/j.jhydrol.2013.12.010, 2014. Wang, W., Xing, W. and Shao, Q.: How large are uncertainties in future projection of reference evapotranspiration through different approaches?, J. Hydrol., 524, 696–700, doi:10.1016/j.jhydrol.2015.03.033, 2015.

Please also note the supplement to this comment:
http://www.hydrol-earth-syst-sci-discuss.net/hess-2015-408/hess-2015-408-AC1-supplement.zip

---

## Author Comment (AC2) · 13 Apr 2016

MS No.: HESS-2015-408R1; MS Type: Research article

We appreciate the thoughtful comments from the reviewers, which have helped us to improve the original manuscript. We explain in detail how we responded to each of the reviewer's comments, with line numbers referring to the revised manuscript unless otherwise noted. We changed our title to "Sensitivity of future Continental United States water deficit projections to General Circulation Model, evapotranspiration estimation method, and greenhouse gas emission scenario" in response to reviewers comment.

In addition, we upload revised manuscript and responses to reviewers as our supplemental material.

[1] Referee review: Before using the GCMs output to force hydrological model (even estimate RET), the some forms of prior bias correction are always conducted due that GCM often show strong bias over historic period (Wood et al., 2002; 2004). I can only believe the authors use the raw data causing I did not find any information associated with the bias correction description in the paper. So how about the matching degree between the GCM-simulated variables and historical observation? And whether some bias correction jobs should be done before employing these GCMs output.

Author's response: We added an explanation in the methods section regarding why we focused on the sensitivity of changes in raw GCM predictions rather than changes in bias-corrected GCM predictions. "Because GCM predictions are known to contain systematic biases (Hwang and Graham, 2013; Wood et al., 2002, 2004) we evaluated the sensitivity of the mean monthly change in raw climate predictions between retrospective and future periods to the choice of GCM, ET0 estimation method and RCP trajectories. This is analogous to using the delta change GCM bias correction method that involves shifting the mean of a series of observed climate data by the mean difference in raw GCM output between the corresponding observed time period and the desired future period. Teutschbein and Seibert (2012) pointed out that all bias correction methods are based on the stationarity principle that assumes that similar biases occur in the retrospective and future predictions and thus the same bias-correction algorithm may be applied to both. Muerth et al. (2013) found that the impact of bias correction on the relative change of flow indicators between retrospective and future periods was weak for most indicators, however Pierce et al. (2015) found that some bias correction methods altered model-projected changes in mean precipitation and temperature. LaFond et al. (2014) found that the delta change GCM bias correction method was more useful for simulating hydrologic extreme events than the quantile mapping bias correction method as it preserved daily climate variability better. In this study, we differenced raw

rather than bias corrected GCM outputs in order to prevent spurious alteration of the climate change signal between retrospective and future GCMs that might be induced by the bias correction method"

[2 and 3] Referee review: GCM simulated temperature is commonly considered to have high confidence than other climatic variables such as vapor pressure and radiation (Randall et al., 2007). The differences of estimated ET between temperature-based ET equations and radiation based equations maybe due to the uncertain input data quality rather than the method selection as the authors declared. In fact, temperature-based equations have been considered not competent in RET change (e.g., Roderick et al., 2009) due that a steady increase in temperature over time will translate into a calculated steady increase in evapotranspiration. Generally, using combination equations maybe more suitable for projection future RET. However, as the above comment pointed out, the GCM-simulated temperature was also widely considered to have relatively high confidence in comparison with other meteorological variables. The different combinations between methods and data should be discussed (see some literatures, Kingston et al., 2009; Wang et al., 2015).

Author's response: The main finding of our paper is that the choice of ET estimation method is as important as GCM selection and the effects of ET estimation method vary depending on region and season. We agree that the effects of the ET estimation method depend both on the physics represented in the method and the reliability of the parameters needed for the method. We revised the manuscript to make this point more clearly and included discussion of the references suggested above on P12: "Kingston et al. (2009) recommended the use of different ET0 equations to evaluate global ET0, and Wang et al. (2015) found that although different methods predict similar future ET0, there are important differences in uncertainties due to ET0 estimation methods and input data reliability. Currently many hydrological models use a single evapotranspiration method for simulation, which may substantially increase the uncertainty and reduce the reliability of future projections. Our results strongly indicate that an ensemble of ET0 estimation methods should be used to understand potential future water availability and water deficit due to climate change."

Furthermore we added a paragraph in the discussion section and a new plot in the supplemental material (Fig. S-3). "GCMs estimate some climate variables, such as temperature, with higher confidence than other variables (Randall et al., 2007). However, for some evapotranspiration estimation methods the effect of temperature on evaporation is smaller than other climate variables ( Linacre, 1994; Thom et al., 1981, Roderick et al., 2009a, 2009b). We found that temperature and net radiation from the CMIP5 GCMs show increasing trends over the 2005-2100 time period, while wind speed and surface pressure are relatively constant (Fig. S-3). Because we considered various ET0 estimation methods our results include the impacts of the different physics represented in the ET0 methods, the projected changes each of the climate variables contributing to the different ET0 methods, and the reliability of the predictions of each variable.

[4] Referee review: ET always mean actual evapotranspiration, it may be better use RET/ET0 to represent reference evapotranspiration.

Author's response: We changed this for clarity and refer to reference evapotranspiration as ET0 throughout the manuscript.

[5 and 6] Referee review: It is better to divide the results into several sub-sections. Results should be presented as such and not mingled with explanations (analysis section), so please separate the results section and discussion section.

Author's response: We divided the previous combined section into separate results and discussion sections.

[Additional Literature cited] Hwang, S. and Graham, W. D.: Development and comparative evaluation of a stochastic analog method to downscale daily GCM precipitation, Hydrol. Earth Syst. Sci., 17(11), 4481–4502, doi:10.5194/hess-17-4481-2013, 2013.

Kingston, D. G., Todd, M. C., Taylor, R. G., Thompson, J. R. and Arnell, N. W.: Uncertainty in the estimation of potential evapotranspiration under climate change, Geophys. Res. Lett., 36(20), L20403, doi:10.1029/2009GL040267, 2009. LaFond, K. M., Griffis, V. W. and Spellman, P.: Forcing Hydrologic Models with GCM Output: Bias Correction vs. the "Delta Change" Method, in World Environmental and Water Resources Congress 2014, vol. 1, pp. 2146–2155, American Society of Civil Engineers, Reston, VA., 2014. Linacre, E. T.: Estimating U.S. Class A Pan Evaporation from Few Climate Data, Water Int., 19(1), 5–14, doi:10.1080/02508069408686189, 1994. Muerth, M. J., Gauvin St-Denis, B., Ricard, S., Velázquez, J. A., Schmid, J., Minville, M., Caya, D., Chaumont, D., Ludwig, R. and Turcotte, R.: On the need for bias correction in regional climate scenarios to assess climate change impacts on river runoff, Hydrol. Earth Syst. Sci., 17(3), 1189–1204, doi:10.5194/hess-17-1189-2013, 2013. Pierce, D. W., Cayan, D. R., Maurer, E. P., Abatzoglou, J. T. and Hegewisch, K. C.: Improved bias correction techniques for hydrological simulations of climate change, J. Hydrometeorol., 150915153707007, doi:10.1175/JHM-D-14-0236.1, 2015. Randall, D. A., Wood, R. A., Bony, S., Colman, R., Fichefet, T., Fyve, J., Kattsov, V., Pitman, A., Shukla, J., Srinivasan, J., Stouffer, R. J., Sumi, A. and Taylor, K. E.: Climate Models and Their Evaluation, in Climate Change 2007: The Physical Science Basis. Contribution of Working Group I to the Fourth Assessment Report of the Intergovernmental Panel on Climate Change, edited by S. Solomon, D. Qin, M. Manning, Z. Chen, M. Marquis, K. B. Averyt, M. Tignor, and H. L. Miller, pp. 591–662, Cambridge University Press, Cambridge, United Kingdom and New York, NY, USA., 2007. Roderick, M. L., Hobbins, M. T. and Farquhar, G. D.: Pan Evaporation Trends and the Terrestrial Water Balance. I. Principles and Observations, Geogr. Compass, 3(2), 746–760, doi:10.1111/j.1749-8198.2008.00213.x, 2009a. Roderick, M. L., Hobbins, M. T. and Farquhar, G. D.: Pan Evaporation Trends and the Terrestrial Water Balance. II. Energy Balance and Interpretation, Geogr. Compass, 3(2), 761–780, doi:10.1111/j.1749-8198.2008.00214.x, 2009b. Teutschbein, C. and Seibert, J.: Bias correction of regional climate model simulations for hydrological climate-change impact studies: Review and evaluation of different methods, J. Hydrol., 456-457, 12–29, doi:10.1016/j.jhydrol.2012.05.052, 2012. Thom, A. S., Thony, J.-L. and Vauclin, M.: On the proper employment of evaporation pans and atmometers in estimating potential transpiration, Q. J. R. Meteorol. Soc., 107(453), 711–736 [online] Available from: http://dx.doi.org/10.1002/qj.49710745316, 1981. Wang, W., Xing, W. and Shao, Q.: How large are uncertainties in future projection of reference evapotranspiration through different approaches?, J. Hydrol., 524, 696–700, doi:10.1016/j.jhydrol.2015.03.033, 2015. Wood, A. W., Leung, L. R., Sridhar, V. and Lettenmaier, D. P.: Hydrologic implications of dynamical and statistical approaches to downscaling climate model outputs, Clim. Change, 62(1-3), 189–216, doi:10.1023/B:CLIM.0000013685.99609.9e, 2004. Wood, A. W., Maurer, E. P., Kumar, A. and Lettenmaier, D. P.: Long-range experimental hydrologic forecasting for the eastern United States, J. Geophys. Res., 107(D20), 4429, doi:10.1029/2001JD000659, 2002.

Please also note the supplement to this comment:
http://www.hydrol-earth-syst-sci-discuss.net/hess-2015-408/hess-2015-408-AC2-supplement.zip

[Figure]

**Fig. 1.** Fig. S-3 Projections of mean maximum temperature, net radiation, wind speed at 2 m surface, and surface pressure of CMIP5 from 2005 to 2010 for RCP 8.5.

---

## Author Response (AR2)

**Author's response letter for "Sensitivity of future Continental United States water deficit projections to General Circulation Model, evapotranspiration estimation method, and greenhouse gas emission scenario." by S. Chang et al.**

**MS No.: hess-2015-408; MS Type: Research article**

We appreciate the thoughtful comments from the referee #2, which have helped us to improve the original manuscript. We explain in detail how we responded to each of comments, with line numbers referring to the revised manuscript unless otherwise noted.

**Referee #2**

| Index | | Comments |
|---|---|---|
| 1 | Referee review | Lines 67-75. Although the authors have summarized some literatures on that future ET0 projections have been shown to depend on ET0 estimation methods in addition to GCMs, here, the authors appear to miss some new literatures on the impact of different combinations between methods and GCM data on projecting ET0 (e.g., Wang et al., 2015). |
| | Author's response | We added the reference in the introduction section. After line 67 : "*For example, Wang et al. (2015) used projections from the CMIP3 HADCM3 model A2 scenario and found that the physically-based Penman-Monteith equation, which uses less reliable GCM projection data (including vapor pressure and solar radiation), and the empirical temperature-based Hargreaves equation showed similar patterns but different magnitudes for future ET0 changes over the Hanjiang River Basin in China.*" |
| 2 | Referee review | Lines 95. Given a large number of publications on future ET0 projection, the motivation of this study should be further explain to emphasize what new contribution it makes to the field, rather than only stated that "adds to the literature". |
| | Author's response | We revised the contribution statement to "*In this study we perform a comprehensive evaluation of the relative sensitivity of future P, ET0 and water deficit (defined here as P- ET0) projections to choice of GCM, ET0 method and RCP trajectory over the continental USA using CMIP5 GCM model outputs to provide new insights that will inform more robust future water resource planning efforts.*" |
| 3 | Referee review | Although the authors divide the results into several paragraphs, it maybe not clear enough for reader. Therefore, it will be better to give the suitable titles for sub-sections |
| | Author's response | We divided the results section into four subsections including: "*3.1. Projected P, ET0, and water deficit change in the 21st century, 3.2. Global sensitivity analysis of projected changes, 3.3. Change in annual mean water deficit projections using different ET0 methods, and 3.4. Monte Carlo filtering.*" |
| 4 | Referee review | I would prefer a shorter conclusion, this one is more like a summary. Some discussion about future research (e.g., Lines 407-413) should be moved to the discussion section. Moreover, the authors need realize that some recent literatures have done the related work on weighting the results of the ensemble of GCMs when projecting ET0 (e.g., Xing et al., 2014). |
| | Author's response | We retitled the section from "*Conclusion*" to "*Summary and Conclusions*", and omitted lines 370-379 from the section. In addition we added the suggested reference (Xing et al. 2014) in the section as the example of weighting the results of an ensemble of GCMs and ET methods. |

| 5 | Referee review | Figure 8 missed the scale of x-axis. |
|---|---|---|
| | Author's response | We replaced figure 8 to include the scale of x-axis. |

Additional literature cited:

[revised manuscript text omitted]